# Flat Minima and Generalization:
# Insights from Stochastic Convex Optimization

**Matan Schliserman** [* 1]   **Shira Vansover-Hager** [* 1]   **Tomer Koren** [1 2]

## Abstract

Understanding the generalization behavior of learning algorithms is a central goal of learning theory. A recently emerging explanation is that learning algorithms are successful in practice because they converge to flat minima, which have been consistently associated with improved generalization performance. In this work, we study the link between flat minima and generalization in the canonical setting of stochastic convex optimization with a non-negative, $\beta$-smooth objective. Our first finding is that, even in this fundamental setting, flat empirical minima may incur trivial $\Omega(1)$ population risk while sharp minima generalizes optimally. We then demonstrate that this phenomenon extends to sharpness-aware algorithms introduced by Foret et al. (2021), namely Sharpness-Aware Gradient Descent (SA-GD) and Sharpness-Aware Minimization (SAM). For SA-GD we prove that it successfully converges to a flat minimum at a fast rate, but the population risk of the solution can still be as large as $\Omega(1)$. For SAM we show that although it minimizes the empirical loss, it may converge to a sharp minimum and also incur population risk $\Omega(1)$. Finally, we establish population risk upper bounds for both SA-GD and SAM using algorithmic stability techniques.

## 1. Introduction

Understanding the generalization behavior of modern learning algorithms has become a central focus of theoretical machine learning. This interest is motivated by the observation that in heavily overparameterized models, the training objective admits many global optima that perfectly fit the data (Zhang et al., 2017); yet, while some of these minimizers generalize poorly, others—typically those to which common optimization algorithms converge—generalize well (Neyshabur et al., 2014; 2017; Zhang et al., 2017). These observations naturally raise the fundamental question of what theoretical and algorithmic conditions ensure that minimizers generalize well.

One prominent condition that has received significant attention is the *flatness* of the minimum. Flat minima, solutions that that remain (approximate) minimizers under small parameter perturbations, have been consistently associated with better generalization, while sharper, non-flat minima are linked with worse out-of-sample performance (Keskar et al., 2016; Dziugaite & Roy, 2018; Jiang et al., 2019; Singh et al., 2025). This insight has motivated a variety of methods that encourage solutions in flat regions of the loss landscape, rather than sharp ones (Wu et al., 2020; Foret et al., 2021; Kwon et al., 2021; Zheng et al., 2021; Du et al., 2021; Kim et al., 2022; Zhuang et al., 2022; Liu et al., 2022; Du et al., 2022; Zhao et al., 2022; Andriushchenko et al., 2023a; Li & Giannakis, 2023; Jiang & Stich, 2023; Xie et al., 2024; Tahmasebi et al., 2024; Li et al., 2024). In particular, Foret et al. (2021) introduced the Sharpness-Aware Minimization (SAM) approach, which reformulates the standard optimization problem as minimizing the *Sharpness-Aware Empirical Risk (SAER)*, defined as $F_S^r(w) = \max_{\|v\| \leq r} F_S(w + v)$ where $F_S$ is the empirical risk over a sample $S$ and $r$ is a *perturbation radius* parameter. This approach encourages solutions robust to parameter perturbations, thus corresponding to flatter minima.

Despite the success of SAM, as well as of other sharpness-aware methods (Bahri et al., 2021; Chen et al., 2021; Foret et al., 2021; Kaddour et al., 2022; Lee et al., 2023), the theoretical link between flatness and generalization remains not fully understood. While some works show that in certain non-convex regimes the flatness of an arbitrary minimizer does not affect generalization (e.g., Dinh et al., 2017; Wen et al., 2023), it is unclear whether this also holds for concrete optimization methods that explicitly aim to find flat minima. For such methods, existing analyses either provide only empirical evidence (Andriushchenko et al., 2023b; Wen

---

[1]Blavatnik School of Computer Science and AI, Tel Aviv University [2]Google Research. Correspondence to: Matan Schliserman <schliserman@mail.tau.ac.il>, Shira Vansover-Hager <shirav@mail.tau.ac.il>.

*Proceedings of the 43ʳᵈ International Conference on Machine Learning*, Seoul, South Korea. PMLR 306, 2026. Copyright 2026 by the author(s).

et al., 2023; Ramasinghe et al., 2023), establish problem parameters-dependent generalization bounds (Neyshabur et al., 2017; Wei & Ma, 2019a;b; Foret et al., 2021; Norton & Royset, 2023), or restrict attention to quadratic or strongly convex objectives (Chen et al., 2024; Tan et al., 2025). As a result, it remains unclear whether and under which conditions finding a flat empirical minimum using such algorithms does in fact lead to improved generalization, or how the generalization guarantees of these practical methods compare to those of standard optimization algorithms such as gradient descent (GD) and stochastic gradient descent (SGD).

In this paper, we aim to gain insight into the relationship between flatness and generalization by studying the above questions within the framework of Stochastic Convex Optimization (SCO): a fundamental and extensively studied theoretical model widely used to analyze stochastic optimization algorithms. SCO is particularly well-suited for such a study, as it is well-known that SCO problems can admit multiple empirical minimizers, not all of which are guaranteed to generalize well (Shalev-Shwartz et al., 2010; Feldman, 2016). We focus on the regime where the loss functions are non-negative and $\beta$-smooth;[1] in this setting, gradient methods such as GD and SGD are known to generalize optimally (Hardt et al., 2016; Nikolakakis et al., 2022), as opposed to the wider convex non-smooth setting (Amir et al., 2021; Schliserman et al., 2025; Livni, 2024; Vansover-Hager et al., 2025). Within this SCO framework, we impose the additional assumption that $f$ admits at least one flat minimum, i.e., a minimizer such that the loss remains constant within a ball of radius $\rho$ around it (we call such a minimizer a $\rho$-flat minimum). To capture this formally, we introduce a strong flatness condition (see Definition 2.1), and analyze the generalization performance of several natural algorithms under this condition.

Our contributions shed light on the extent to which flatness relates to generalization in SCO. We construct examples showing that flat empirical minima can generalize poorly, demonstrating that minimizing the Sharpness-Aware Empirical Risk (SAER) does not in itself guarantee good generalization. First, we present an SCO instance in which there exists a flat empirical risk minimizer (ERM) that generalizes poorly, while within the same setting, there exists a sharp ERM that generalizes well. Then, we show that this poor generalization behavior extends to two natural "sharpness-aware" algorithms originally proposed by Foret et al. (2021), designed to bias optimization toward flat solutions: Sharpness-Aware Gradient Descent (SA-GD)[2] and

---

[1]A differentiable function $f : \mathbb{R}^d \to \mathbb{R}$ is $\beta$-smooth if $\|\nabla f(v) - \nabla f(u)\|_2 \le \beta \|v - u\|_2$ for all $u, v \in \mathbb{R}^d$.

[2]This algorithm was introduced in (Foret et al., 2021) without being explicitly named, referred here as SA-GD for conciseness.

Sharpness-Aware Minimization (SAM). For SA-GD, we prove that it indeed converges to a flat minimum, however, there are instances where it converges to solutions that generalize strictly worse compared to those found by standard GD and SGD, which are known to generalize optimally in the same setting. These results indicate that even flat minima found algorithmically using a sharpness-aware gradient method might generalize poorly. For SAM we observe a sharper contrast: although it minimizes the empirical risk, it does not necessarily minimize sharpness as it may converge to a non-flat minimum, and similarly to SA-GD, we show it might converge to minima with poor generalization compared to (S)GD. These results provide insight into possible limitations of sharpness-aware approaches in terms of the flatness of the solution found and its out-of-sample performance relative to (S)GD. Finally, we derive new population loss upper bounds for SA-GD and SAM. Compared to (S)GD, these bounds include an additional dominant term that nearly matches our lower bounds.

## 1.1. Summary of contributions

In more detail, we make the following technical contributions. (The bounds presented below describe the dependence on the number of iterations $T$, number of training examples $n$, step size $\eta$, smoothness parameter $\beta$, flatness radius of the loss minimizer $\rho$, and perturbation size $r$.)

(i) We introduce a strong flatness condition assuming the existence of a perfectly flat minimum of radius $\rho$. For Sharpness-Aware ERM (SA-ERM), even under this strong condition, we construct a smooth SCO problem where the empirical risk admits a flat minimizer with population risk $\Omega(1)$, while a non-flat minimizer achieves optimal generalization (Theorem 3.1).

(ii) For the SA-GD algorithm (Foret et al., 2021), we prove an empirical optimization bound $O(1/\eta T + \max(r - \rho, 0)^2)$, implying that with $\eta \simeq 1/\beta$ and $r \simeq \rho$, SA-GD converges to a $\Theta(\rho)$-flat minimum at rate $O(1/T)$. In contrast, we establish a lower bound of $\Omega(\eta^2(r-\rho)^2 T)$ on the population loss of SA-GD for $r \gtrsim \rho$, showing that SA-GD may generalize poorly even when converging to flat minima. In particular, tuning the algorithm with $\eta \simeq 1/\beta$ and $r \gtrsim \rho + 1/\sqrt{T}$ can lead to a population risk of $\Omega(1)$ (Theorems 4.1 and 4.3). Finally, using algorithmic stability, we prove a population upper bound for SA-GD under $\rho$-flatness (Theorem 4.4). This bound nearly matches our lower bound, and compared to vanilla GD and SGD it contains an additional dominant term $O(\eta^2 r^2 T)$.

(iii) For SAM (Foret et al., 2021), we obtain the same bound $O(1/(\eta T) + \max(r - \rho, 0)^2)$ for the empirical risk, but also show a convex, smooth case where SAM converges to a sharp minimum, i.e., it fails to minimize

the SAER. As for generalization, we establish a population lower bound of $\Omega(\eta^2 r^2 T)$ in the case $\rho = 0$, which implies a trivial risk of $\Omega(1)$ when $\eta \simeq 1/\beta$ and $r \gtrsim 1/\sqrt{T}$, or when $\eta \simeq 1/\sqrt{T}$ and $r = \Theta(1)$, regimes where SAM minimizes the empirical risk (Theorems 5.1 to 5.3). As with SA-GD, we prove a population upper bound for SAM under $\rho$-flatness, achieving the same rate as SA-GD (Theorem 5.4).

To our knowledge, these results are the first to formally address the connection between flatness and generalization in the convex regime, and they bear some interesting implications. On the positive side, they provide the first indication that sharpness-aware methods converge at a dimension-independent fast $O(1/T)$ rate in terms of empirical risk for general convex optimization, despite the SAER objective being non-smooth, and this convergence can further benefit from flatness of the objective. On the negative side, our results show that even in the basic convex and smooth regime, a sharp empirical minimum may generalize better than a flat one, and this can occur when the flat empirical risk minimizer is selected arbitrarily, e.g., by the SA-ERM algorithm, or algorithmically, by the SA-GD algorithm. Furthermore, our findings highlight that optimization methods explicitly designed to locate flat minima, such as SA-GD and SAM, may converge to solutions that generalize poorly. In contrast, standard gradient-based methods like GD and SGD are known to achieve optimal generalization in this setting when using the optimization-optimal step size $\eta \simeq 1/\beta$ (Lei & Ying, 2020; Nikolakakis et al., 2022).

## 1.2. Related work

**Flat minima and generalization.** The conjectured connection between flat minima and generalization dates back to Hochreiter & Schmidhuber (1997). Since then, a large body of empirical and theoretical work has suggested that flatter minima correlate with, or even guarantee, better generalization performance (Keskar et al., 2016; Dziugaite & Roy, 2017; Neyshabur et al., 2017; Wu et al., 2018; Jastrzebski et al., 2018; Jiang et al., 2019; Wei & Ma, 2019a;b; Blanc et al., 2020; HaoChen et al., 2021; Foret et al., 2021; Damian et al., 2021; Li et al., 2021; Ma & Ying, 2021; Nacson et al., 2022; Wei et al., 2022; Lyu et al., 2022; Norton & Royset, 2023; Wu & Su, 2023; Ding et al., 2024). However, several works caution against interpreting flatness as a universal predictor of generalization (Dinh et al., 2017; Andriushchenko et al., 2023b; Wen et al., 2023; Ramasinghe et al., 2023). Notably, from a theoretical perspective, Dinh et al. (2017) showed that in ReLU networks sharpness can be arbitrarily altered through reparameterization without affecting the learned function or its generalization, implying that common flatness measures are not parameterization-invariant and may therefore be misleading.

More recently, Wen et al. (2023) examined two-layer ReLU networks defining flatness as the trace of the Hessian. Using this architecture and notion of flatness they identified scenarios where flat minima fail to generalize, while sharpness-minimization algorithms such as SAM may still succeed, although their analysis of SAM was only empirical. Our results go beyond both works: unlike (Dinh et al., 2017), we give explicit constructions where flat minimizers fail while sharp minimizers generalize perfectly, directly challenging the conjecture itself, and unlike (Wen et al., 2023), we establish this phenomenon already in the fundamental convex $\beta$-smooth setting and under much stronger flatness assumptions. Furthermore we provide theoretically provable lower bounds on the generalization of SAM, offering a more rigorous understanding of its limitations.

**Convergence rates of SAM.** Many works on the convergence of SAM analyze a variant of SAM that does not use gradient normalization during the ascent step (Andriushchenko & Flammarion, 2022; Behdin & Mazumder, 2023; Agarwala & Dauphin, 2023; Kim et al., 2023). This variant does not match practical implementations of SAM, where normalization is typically used (Si & Yun, 2023), and more recent work showed that normalization improves SAM's performance (Dai et al., 2023). Our work considers SAM with normalization and provides more practical bounds. Another line of research studies the implicit bias of SAM and its variants under infinitesimal step sizes (Wen et al., 2022; Andriushchenko & Flammarion, 2022), while we focus on the practical discrete setting.

In more specific cases, Bartlett et al. (2023) gave convergence rates for SAM on convex quadratics, whereas our work addresses general smooth convex objectives. Recent works also consider smooth nonconvex objectives with decaying or sufficiently small $r$ (Mi et al., 2022; Zhuang et al., 2022; Sun et al., 2024), but such assumptions differ from practice, where $r$ might be a constant. Our bounds instead cover smooth convex functions and hold for any $r$, including large values. Finally, Si & Yun (2023) derived convergence guarantees in deterministic and stochastic regimes, but in the smooth convex case they only proved convergence to stationary points, leaving convergence to global minima as an open problem. We close this gap by providing the first rates of convergence to global minima for SAM on general smooth convex objectives, and we are the first to incorporate the true flatness of the objective into the convergence analysis.

**Generalization of SAM.** Foret et al. (2021), who originally introduced SAM, established PAC-Bayes bounds to explain its generalization. These bounds are dimension dependent and may be vacuous in many scenarios. More recently, Tan et al. (2025) analyzed the smooth and strongly

convex setting, comparing the algorithmic stability of SAM and SGD. Chen et al. (2024) studied generalization from a different angle, comparing the conditions for benign overfitting under SGD and SAM in two-layer convolutional ReLU networks. In contrast to these works, we establish the first dimension-independent generalization bounds for the broad class of smooth convex (but not strongly convex) objectives, together with the first lower bounds on the generalization performance of SAM in this setting.

**Generalization in SCO.** Stochastic convex optimization is a fundamental theoretical framework for analyzing widely used optimization algorithms, where the loss function is assumed to be convex and Lipschitz. In this setting, prior work (Shalev-Shwartz et al., 2010; Feldman, 2016; Carmon et al., 2023) have shown that, although learning in this framework is possible (e.g., via Stochastic Gradient Descent), empirical risk minimization (ERM) may fail (even under additional assumptions such as smoothness and realizability), since uniform convergence does not generally hold. In our work, we focus on flat ERMs, namely minimizers of the SAER, and demonstrate that even when the minima are flat, they may still generalize poorly. Beyond ERM, several natural algorithms such as full-batch Gradient Descent and multi-pass Stochastic Gradient Descent have also been shown to fail in this setting (Amir et al., 2021; Livni, 2024; Schliserman et al., 2025; Vansover-Hager et al., 2025). All of these works focus on the non-smooth regime and establish lower bounds in that setting. In contrast, our work studies the generalization of Sharpness-Aware Minimization algorithms in smooth and realizable SCO, and we show that even under these strong assumptions, SA-GD and SAM may still generalize poorly.

**Smooth SCO with low noise.** The problem of smooth stochastic convex optimization with low noise as been extensively studied. (Srebro et al., 2010) established that Stochastic Gradient Descent (SGD) attains a risk bound of $O(1/n)$ in this setting. This result was recently extended by (Attia et al., 2025) to the last iterate of SGD. In our work, we demonstrate that in the deterministic setting, SA-GD and SAM also attain these optimal rates when applied to smooth loss functions. In addition, for SA-GD we prove an even stronger result: under an additional flatness condition, the method achieves the same fast rates for convergence with respect to the SAER $F_S^r$, a function that is generally non-smooth. From a generalization perspective, recent work (Lei & Ying, 2020; Nikolakakis et al., 2022; Schliserman & Koren, 2022; Evron et al., 2026; Attia et al., 2025) has used stability arguments to show that gradient methods such as GD and SGD, both with and without replacement and with $T = n$, achieve an optimal risk of $O(1/n)$ in this setting. Our work shows that, in contrast to those algorithms, SA-GD and SAM may generalize poorly, even in smooth

and realizable SCO.

## 2. Problem setup

We study the generalization properties of flat minima in the framework of (smooth) *Stochastic Convex Optimization* (SCO). In this setting, there exists a population distribution $\mathcal{D}$ over an instance space $\mathcal{Z}$, and a loss function $f : W \times \mathcal{Z} \to \mathbb{R}$ defined on a convex domain $W \subseteq \mathbb{R}^d$. For any fixed instance $z \in \mathcal{Z}$, the function $f(\cdot, z)$ is assumed to be non-negative, convex, and $\beta$-smooth ($\beta > 0$) with respect to its first argument $w$. The learning goal is to minimize the *population risk*, defined as the expected loss over $\mathcal{D}$,

$$F(w) := \mathbb{E}_{z \sim \mathcal{D}}[f(w, z)]. \qquad (1)$$

Since $\mathcal{D}$ is unknown, learning algorithms instead use a finite i.i.d. sample $S = \{z_1, \ldots, z_n\}$ drawn from $\mathcal{D}$. A common approach is to minimize the *empirical risk* over $S$, given by

$$F_S(w) := \frac{1}{n} \sum_{i=1}^{n} f(w, z_i). \qquad (2)$$

A main focus of this paper is on objective functions that admit *flat minima*, formalized as follows.

**Definition 2.1** ($\rho$-flatness). We say that $w^\star \in \mathbb{R}^d$ is a *$\rho$-flat minimum* (for $\rho \geq 0$) of a non-negative function $f : \mathbb{R}^d \to \mathbb{R}$ if for every $w \in \mathbb{R}^d$ with $\|w - w^\star\| \leq \rho$, it holds that $f(w) = 0$. If such a $\rho$-flat minimum exists for $f$, we also say that $f$ is $\rho$-flat; the maximal $\rho$ satisfying this condition is called the *flatness radius* of $f$.

Note that this is a very strong notion of flatness: it in particular implies that the empirical minimization problem with a $\rho$-flat $F_S$ is *realizable* (i.e., there exists $w^\star$ such that $f(w^\star, z_i) = 0$ for almost all $z_i \in S$) and further that $F_S$ is *perfectly flat* in a neighborhood of $w^\star$. Since our goal is to understand the relationship between flatness and generalization, we find it more informative to analyze this connection under the most stringent and unambiguous condition of flatness. In particular, imposing such a condition makes any negative results (i.e., lower bounds) only *stronger*, since they hold even under the most favorable notion of flatness.

With the above notion of flatness in mind, we focus on three natural algorithms:

- *Sharpness-Aware Empirical Risk Minimization (SA-ERM).* The first (meta-)algorithm is a natural, "Sharpness-Aware" variant of ERM that computes, given a parameter $r > 0$:

$$
\begin{aligned}
w_S &\in \arg\min_{w \in W} F_S^r(w), \\
\text{where} \quad F_S^r(w) &= \max_{v:\, \|v\| \leq r} F_S(w + v).
\end{aligned}
\qquad (3)
$$

Namely, it outputs a minimizer of the *sharpness-aware empirical risk* (SAER) with radius $r$, which we denote

by $F_S^r$. The idea here is that, if the empirical risk $F_S$ is $\rho$-flat and $r \le \rho$, then *any* minimizer of the SAER is also a $r$-flat minimum of the original empirical risk $F_S$.

- ***Sharpness-Aware Gradient Descent (SA-GD).*** The second algorithm is a first-order instantiation of SA-ERM, proposed in (Foret et al., 2021), obtained by running gradient descent on the SAER objective. Starting from $w_1 \in W$ and given parameters $\eta, r > 0$, it takes steps for $t = 1, \ldots, T$ of the form:

$$w_{t+1} = w_t - \eta \nabla F_S(w_t + v_t),$$
$$\text{where} \quad v_t \in \arg \max_{v: \|v\| \le r} F_S(w_t + v). \qquad (4)$$

- ***Sharpness-Aware Minimization (SAM).*** The third algorithm is the original SAM algorithm proposed in (Foret et al., 2021) as a computationally efficient approximation of SA-GD. SAM circumvents the explicit maximization over $v$ in Eq. (4) by replacing $v_t$ with the normalized gradient at $w_t$. Thus, starting from $w_1 \in W$ and given $\eta, r > 0$, the updates of SAM for $t = 1, \ldots, T$ take the form

$$w_{t+1} = w_t - \eta \nabla F_S\left(w_t + r \frac{\nabla F_S(w_t)}{\|\nabla F_S(w_t)\|}\right). \qquad (5)$$

**Notations.** We denote by $\|\cdot\|$ the $\ell_2$ norm. The symbol $\odot$ represents element-wise multiplication, i.e., $(x \odot y)(i) = x(i)\, y(i)$. Finally, we write $[x]_+$ for the element-wise ReLU function, defined as $[x]_+(i) = \max\{x(i), 0\}$.

## 3. SA-ERM: Generic flat minima

We begin by establishing a lower bound on the generalization performance of SA-ERM. In particular, we construct an SCO instance where, with constant probability, there exists a minimizer of the SAER with a trivial $\Omega(1)$ population risk. This result illustrates not only the limitations of the SA-ERM algorithm in the general smooth SCO setting but also how the loss landscape affects generalization. The result is formalized in the following theorem.

**Theorem 3.1.** *For every $n \in \mathbb{N}$ and $0 \le \rho \le \frac{1}{2}$, let $d = 2^n + 1$ and define $W = \{x \in \mathbb{R}^d : \|x\| \le 1\}$. Then there exist an instance set $\mathcal{Z}$, a distribution $\mathcal{D}$ over $\mathcal{Z}$, and a loss function $f : W \times \mathcal{Z} \to \mathbb{R}$ that is convex, 1-Lipschitz, 1-smooth and $\rho$-flat, such that with probability at least $\frac{1}{2}$ over the training set $S$, there exist $w^{(1)}, w^{(2)} \in \arg \min_{w \in W} F_S(w)$ satisfying:*

- *(i) for every $r \ge 0$, it holds that $w^{(1)} \in \arg \min_{w \in W} F_S^r(w)$. In particular, if $r \le \rho$ then $w^{(1)}$ is an $r$-flat minimum of $F_S$;*

- *(ii) $w^{(2)}$ is a sharp minimum, in the sense that $F_S^\delta(w^{(2)}) \ge F_S(w^{(2)}) + \frac{1}{2}\delta^2$ for all $\delta > 0$.[3]*

---

[3]This condition means that in every neighborhood of the min-

*(iii) we have $F(w^{(1)}) - F(w^\star) = \Omega(1)$, while $F(w^{(2)}) - F(w^\star) = 0$.*

Theorem 3.1 indicates that even when the loss is convex and $\beta$-smooth, and under the arguably strongest notion of flatness (Definition 2.1), a flat minimum of the empirical risk may generalize poorly, whereas a sharp minimum of the same function can generalize optimally. We provide here a proof sketch, the full prove is deferred to Appendix A.

*Proof sketch.* Our construction builds on classical lower bounds in stochastic convex optimization showing the existence of an empirical risk minimizers that overfits (Shalev-Shwartz et al., 2010; Feldman, 2016). In particular, Shalev-Shwartz et al. (2010) consider an instance space $Z = \{0, 1\}^d$ with $d = 2^n$, where examples are drawn uniformly at random. Their loss function is of the form:

$$g(w, z) = \frac{1}{2} \sum_{i=1}^{d} z(i)w(i)^2.$$

With high probability, there exists a coordinate $I$ such that all sampled examples satisfy $z(I) = 0$. The corresponding basis vector $e_I$ is then an ERM but incurs large population loss, yielding a spurious empirical minimizer.

A key difficulty in extending this construction is that the spurious ERM is not a flat minimizer, whereas SA-ERM favors flat solutions. To address this, we use the observation that the function $h : \mathbb{R} \to \mathbb{R}$ defined by

$$h(x) = \frac{1}{2} \max(x - \rho, 0)^2$$

is 1-smooth and $\rho$-flat around its minimizers for $\rho \le \frac{1}{2}$. Composing this function with a suitable variant of the above construction yields a smooth SCO instance in which SA-ERM generalizes poorly. The resulting function is as follows:

$$f(w, z) = \frac{1}{2} \left[ \sqrt{\sum_{i=1}^{d} z(i)w(i)^2 + w(d+1)^2} - \rho \right]_+^2.$$

$\square$

## 4. SA-GD: Algorithmically chosen flat minima

In the previous section, we presented in Theorem 3.1 a hard instance showing that an existence of a flat minimizer that generalizes poorly. However, the same instance also admits flat minima that generalize well. This raises a natural question: does this failure extend to practical algorithms

---

imizer there exists a point with large $F_S$. The inequality is the tightest possible: due to 1-smoothness, any minimizer $w^\star$ of $F_S$ satisfies $F_S^\delta(w^\star) \le F_S(w^\star) + \frac{1}{2}\delta^2$ for all $\delta > 0$.

explicitly designed to seek flat minima, such as SA-GD, or do such methods tend to converge to "good" flat minima? In this section we show that the former holds, and that the lower bound from Theorem 3.1 also applies to flat minima obtained by SA-GD.

We first establish a theorem on the optimization error of SA-GD, showing that when the perturbation radius $r$ is properly tuned, SA-GD minimizes the SAER objective and converges to a flat minimum of the empirical risk. The result is formalized in the following theorem whose proof is deferred to Appendix B.1.

**Theorem 4.1.** *Assume that $f(w, z)$ is $\beta$-smooth, convex, non-negative and $\rho$-flat for all $z$. Let $\{w_t\}_{t=1}^T$ be produced by SA-GD for $T$ steps (Eq. (4)) with $\eta \leq 1/4\beta$ and $r > 0$. For $\widehat{w} := \frac{1}{T}\sum_{t=1}^T, w_t$ it holds that*

$$F_S(\widehat{w}) \leq F_S^r(\widehat{w}) \leq \frac{\|w_1 - w^\star\|^2}{\eta T} + 4\beta \max\{r - \rho, 0\}^2.$$

*In particular, when $\eta = 1/4\beta$, $\|w_1 - w^\star\| = O(1)$ and $r - \rho = O(1/\sqrt{T})$, it holds that*

$$F_S(\widehat{w}) \leq F_S^r(\widehat{w}) \leq O(\beta/T).$$

Theorem 4.1 highlights the effect of flatness on the convergence rate of the algorithm. When the flatness radius $\rho$ is small, the algorithm incurs an additive $O(r^2)$ term in the bound on the SAER objective. In contrast, when $\rho$ is large, even for $r \approx \rho$, SA-GD still minimizes the SAER objective and converges to a flat empirical minimum. Moreover, although SA-GD can be viewed as gradient descent applied to a potentially non-smooth function,[4] its convergence rate in this case matches that of gradient descent on smooth functions.

For the proof, we first make use of the following key lemma, which establishes a regret bound for general algorithms whose update rule takes the form $w_{t+1} = w_t - \eta\nabla F_S(w_t + v_t)$, for $\|v_t\| \leq r$.

**Lemma 4.2.** *Assume that for every $z$, $f(w, z)$ is $\beta$-smooth, convex, non-negative and $\rho$-flat. Let $A$ be an algorithm that given a data set $S$, produces a sequence $\{w_t\}_{t=1}^T$ such that $w_{t+1} = w_t - \eta\nabla F_S(w_t + v_t)$, where $\{v_t\}_{t=1}^T$ are vectors such that for every $t$, $\|v_t\| \leq r$ and $\eta \leq 1/4\beta$. It holds that,*

$$\frac{1}{T}\sum_{i=1}^T F_S(w_t + v_t) - F_S(w^\star)$$

$$\leq \frac{\|w_1 - w^\star\|^2}{\eta T} + 4\beta \max\{r - \rho, 0\}^2.$$

---

[4]For example, if $F_S(x) = x^2$, which is $\beta$-smooth for $\beta = 2$, then $F_S^r(x) = (|x| + r)^2$ is non-smooth.

Next, we show that even when SA-GD converges to a flat minimum, the resulting solution is not guaranteed to generalize well. To demonstrate this, we establish the following lower bound on the population risk of SA-GD.

**Theorem 4.3.** *For every $n, T \in \mathbb{N}, \eta > 0, r \geq 0, \rho < r\left(1 - \frac{3}{3+\eta\sqrt{T}}\right)$, assume $\eta(r-\rho) \leq \frac{1}{\sqrt{T}}$, let $d = 2^n T$ and define $W = \{x \in \mathbb{R}^d : \|x\| \leq 1\}$. Then there exists an instance set $\mathcal{Z}$, a distribution $\mathcal{D}$ over $\mathcal{Z}$, function $f : W \times \mathcal{Z} \to \mathbb{R}$ that is convex $1$-smooth, $1$-Lipschitz and $\rho$-flat, such that for a training set $S$ it holds that with probability at least $\frac{1}{2}$, running SA-GD for $T$ steps yields for every $\tau \in [T]$ suffix average $\widehat{w}_\tau = \frac{1}{T-\tau+1}\sum_{t=\tau}^T w_t$:*

$$F(\widehat{w}_\tau) - F(w^\star) = \Omega(\eta^2(r - \rho)^2 T).$$

In particular, it follows that for step size $\eta \approx 1/\beta$ and perturbation radius $r \approx \rho + 1/\sqrt{T}$, the population risk of SA-GD can be as high as $\Omega(1)$, despite converging to a flat empirical minimum, as shown in Theorem 4.1. This result extends the poor generalization result of flat minima given in Theorem 3.1 also to SA-ERMs that is chosen algorithmically by a natural sharpness-aware gradient method. We provide here a proof sketch, the full proof is deferred to Appendix B.2.

*Proof sketch.* The main technical challenge in the proof is that, in the non-smooth setting, prior constructions (e.g., (Amir et al., 2021; Koren et al., 2022; Livni, 2024; Schlisserman et al., 2025; Vansover-Hager et al., 2025)) exploit non-smoothness to shape the algorithm's dynamics, whereas in the smooth setting such an approach is not possible. Instead, our key idea is to control the sequence of maximizers

$$\{v_t \in \arg\max_{\|v\| \leq r} F_S(w_t + v)\}_{t=1}^T$$

to direct the dynamics toward a spurious ERM, and make sure that the sequence $\{v_t\}_{t=1}^T$ are aligned to such directions. For this, we base our hard instance on the construction for SA-ERM given in Theorem 3.1. In that construction, in the first iteration we have $v_1 = re_i$, where $e_i$ corresponds to the spurious ERM. As a result, SA-GD makes a single step of size $\eta(r - \rho)$ toward this bad ERM. The remaining challenge is to ensure that the algorithm takes $T$ such steps in this direction. To achieve this, we replicate the construction across $T$ mutually orthogonal subspaces, writing $w = (w^{(1)}, \ldots, w^{(T)})$ with each block containing an independent copy of the hard instance. In this way, since $v_t$ is chosen in a different subspace at each iteration $t$, the algorithm makes a single step in each subspace and eventually converges to a bad ERM. The resulting loss function is:

$$f(w, z) = \frac{1}{2}\left[\sqrt{\sum_{i=1}^{2^n}\sum_{t=1}^T [z(i)w^{(t)}(i)]_+^2 + w(d)^2} - \rho\right]_+^2.$$

$\square$

Finally, we establish an upper bound for the population loss achieved by SA-GD in the following theorem.

**Theorem 4.4.** *For every $z$, assume that $f(w, z)$ is $\beta$-smooth, convex, non-negative and $\rho$-flat. Let $\{w_t\}_{t=1}^{T}$ be produced by SA-GD for $T$ steps (Eq. (4)) with $\eta \leq 1/4\beta$ and $r > 0$. For $\widehat{w} := \frac{1}{T} \sum_{t=1}^{T} w_t$, it holds that*

$$\mathbb{E} F(\widehat{w}) \leq O\Bigg[ \frac{\|w_1 - w^\star\|^2}{\eta T} + \eta \beta^2 r^2 T + \frac{\beta^2 \eta T}{n^2}$$
$$+ \left( \beta + \frac{\beta^3 \eta^2 T^2}{n^2} \right) \max\{r - \rho, 0\}^2 \Bigg].$$

*In particular for $T = n, \eta = O(1/\beta)$, $\|w_1 - w^\star\| = O(1)$ and $r - \rho = O(1/\sqrt{T})$ it holds that,*

$$\mathbb{E} F(\widehat{w}) = O\left( \frac{\beta}{n} + \beta r^2 n \right).$$

We note that when $r = 0$, the bound in Theorem 4.4 coincides with the risk bounds of Nikolakakis et al. (2022); Lei & Ying (2020) for GD and SGD in convex, smooth, realizable settings. However, when $r > 0$, our bound contains an additional excess term of $\eta \beta^2 r^2 T$ compared to GD and SGD. This term nearly matches our lower bound in Theorem 4.3, up to its dependence on $\eta$ and $\rho$.

The proof is deferred to Appendix B.3 and relies on a leave-one-out algorithmic stability argument. Specifically, we show that replacing a single training sample results in only a small change in the learned model, which in turn implies a small change in the loss. This stability property allows us to bound the generalization gap, that is, the difference between the empirical risk and the population loss (Bousquet & Elisseeff, 2002; Hardt et al., 2016).

## 5. SAM: Practically chosen flat minima

Finally, we analyze SAM, a well-studied and practically relevant algorithm introduced by (Foret et al., 2021) as a computationally efficient approximation of SA-GD. For this algorithm, we establish the following bound on the empirical risk. Similarly to SA-GD, the flatness of the empirical risk plays a significant role in the convergence of SAM, achieving fast convergence rates the function is $\rho$-flat and $r \leq \rho + 1/\sqrt{T}$.

**Theorem 5.1.** *For every $z$, assume that $f(w, z)$ is $\beta$-smooth, convex, non-negative and $\rho$-flat. Let $\{w_t\}_{t=1}^{T}$ be produced by SAM for $T$ steps (Eq. (4)) with $\eta \leq 1/4\beta$ and $r > 0$. For $\widehat{w} := \frac{1}{T} \sum_{t=1}^{T} w_t$, it holds that*

$$F_S(\widehat{w}) \leq \frac{\|w_1 - w^\star\|^2}{\eta T} + 4\beta \max\{r - \rho, 0\}^2.$$

*In particular, if $\eta = \frac{1}{4\beta}$, $\|w_1 - w^\star\| = O(1), r - \rho = O(1/\sqrt{T})$,*

$$F_S(\widehat{w}) \leq O(\beta/T).$$

The proof is deferred to Appendix C.1. We note that Theorem 5.1 establishes convergence rates in terms of the empirical risk. A natural question is whether SAM achieves similar rates for the SAER. In the following theorem, we show that this is not the case: SAM might incur an additional term of $\Omega(r^2)$ in the convergence rate for the SAER, even for $\rho$-flat functions. This demonstrates that SAM can converge to a non-flat minimum, even when a $\rho$-flat minimum exists.

**Theorem 5.2.** *For every $\eta > 0, n \in \mathbb{N}, r, \rho \leq \frac{1}{2}, W = [-1, 1]$ there exists an instance set $\mathcal{Z}$ and a loss function $F_S : W \times \mathcal{Z} \to \mathbb{R}$ that is non-negative, convex, 1-Lipschitz, 1-smooth and $\rho$-flat such that running SAM on $F_S$ for $T$ steps holds, for any suffix average $\widehat{w}_\tau = \frac{1}{T-\tau+1} \sum_{t=\tau}^{T} w_t$,*

$$\forall \, 0 \leq r \leq \tfrac{1}{2}: \qquad F_S^r(\widehat{w}_\tau) - F_S^r(w^\star) = \Omega(r^2),$$

*that is, SAM converges to a sharp minimum.*

The proof of Theorem 5.2 is deferred to Appendix C.2.

Finally, we turn to discuss the generalization guarantees of SAM. In the following lower bound, we show that SAM can exhibit poor generalization in SCO under the realizable setting ($\rho = 0$), leaving the $\rho$-flat case ($\rho \gg 0$) for future work.

**Theorem 5.3.** *Given $n \geq 6, T \geq 6, \eta, r > 0$ such that $\eta r \leq 1/2\sqrt{T}$, let $d = 2^n T$ and $W = \{w \in \mathbb{R}^d : \|w\| \leq 1\}$. Then there exists an instance set $\mathcal{Z}$, a distribution $\mathcal{D}$ over $\mathcal{Z}$, a convex 6-smooth 6-Lipschitz and realizable function $f : W \times \mathcal{Z} \to \mathbb{R}$ such that for a training set $S$ with probability at least $\frac{1}{3}$ running SAM for $T$ steps with trajectory $\{w_t\}_{t=1}^{T}$, yields for every $\tau \in [T]$ suffix average $\widehat{w}_\tau = \frac{1}{T-\tau+1} \sum_{t=\tau}^{T} w_t$:*

$$F(\widehat{w}_\tau) - F(w^\star) = \Omega(\eta^2 r^2 T).$$

We provide here a proof sketch, the full proof is deferred to Appendix C.3.

*Proof sketch.* As with the construction for SA-GD in Theorem 4.3 we begin with a loss that admits a spurious ERM and $T$ orthogonal subspaces,

$$f_1(w, z) = \frac{1}{2} \sum_{i=1}^{2^n} \sum_{t=2}^{T} z(i) \, w^{(i)}(t)^2.$$

The main difficulty in this context is that the algorithm is initialized at $w_1 = 0$, which, in previous constructions, is already a minimizer of the empirical risk. As a result, if we were to apply the same approach, SAM would remain at initialization throughout training and thus generalize well.

To overcome this challenge, our key idea is to exploit the normalization of the ascent step, which can amplify small

perturbations into meaningful progress. We begin by introducing a sufficiently small linear loss in the first orthogonal subspace,

$$f_2(w, z) = \frac{\gamma}{2} \left[ v_z^\top w^{(1)} + \delta_1 \right]_+^2,$$

for an appropriate choice of $v_z$ depending on the samples $z$, with small $\delta_1, \gamma > 0$. Although the gradients of this function at the initialization point are small, the normalization step amplifies them, producing a progress of $\eta r$ toward the bad ERM in this subspace. To extend this effect across $T$ orthogonal subspaces, we design a chaining mechanism that couples consecutive subspaces, such that progress in one subspace activates progress in the next. This is achieved via the following function:

$$f_3(w) = \frac{1}{2} \sum_{i=1}^{2^n} \sum_{t=2}^{T} \left[ w^{(i)}(t) - \delta_t \cdot w^{(i)}(t-1), 0 \right]_+^2,$$

for an appropriate choice of $\{\delta_t\}_{t=2}^{T}$. This construction ensures that the algorithm makes progress of order $\eta r$ in each of $T$ distinct subspaces, ultimately guiding the iterates toward a spurious ERM that generalizes poorly.

The final loss function is therefore

$$f(w, z) = f_1(w, z) + f_2(w) + f_3(w, z). \qquad \square$$

Finally, we establish an upper bound for the population loss achieved by SAM in the following theorem.

**Theorem 5.4.** *For every $z$, assume that $f(w, z)$ is $\beta$-smooth, convex, non-negative and $\rho$-flat. Let $\{w_t\}_{t=1}^{T}$ be produced by SAM for $T$ steps (Eq. (5)) with $\eta \leq 1/4\beta$ and $r > 0$. For $\widehat{w} := \frac{1}{T} \sum_{t=1}^{T} w_t$, it holds that*

$$\mathbb{E}F(\widehat{w}) \leq O \left[ \frac{\|w_1 - w^\star\|^2}{\eta T} + \eta \beta^2 r^2 T + \frac{\beta^2 \eta T}{n^2} \right.$$
$$\left. + \left( \beta + \frac{\beta^3 \eta^2 T^2}{n^2} \right) \max\{r - \rho, 0\}^2 \right].$$

*In particular for $T = n, \eta = O(1/\beta)$, $\|w_1 - w^\star\| = O(1)$ and $r - \rho = O(1/\sqrt{T})$ it holds that,*

$$\mathbb{E}F(\widehat{w}) = O \left( \frac{\beta}{n} + \beta r^2 n \right).$$

As in the bound for SA-GD, we note that when $r = 0$, the bound in Theorem 4.4 coincides with the risk bounds of Lei & Ying (2020); Nikolakakis et al. (2022) for GD and SGD in convex, smooth, realizable settings. However, when $r > 0$, our bound contains an additional excess term of $\eta \beta^2 r^2 T$ relative to GD and SGD. This term nearly matches our lower bound in Theorem 5.3, up to its dependence on $\eta$.

The proof follows from an algorithmic stability analysis, similar to that in Theorem 4.4;, and is given in Appendix C.4.

## 6. Discussion and Limitations

In this work, we study the relationship between flat minima and generalization. We focus on the fundamental convex and smooth setting and provide the first upper and lower bounds on both optimization and generalization for three natural, extensively studied sharpness-aware methods: SA-ERM, SA-GD, and SAM. To the best of our knowledge, our work provides the first provable separation showing that, even in convex problems, sharpness-aware algorithms can exhibit worse generalization than standard GD and SGD, and that explicitly seeking flat minima does not necessarily improve performance and can, in fact, lead to worse outcomes.

**Limitations.** It is important to note that our separation between GD and SGD versus SA-GD and SAM is shown in a worst case setting. It relies on a specific loss function, data distribution, gradient oracle, and particular hyperparameter choices such as large step sizes. This does not mean that sharpness aware methods always perform worse than GD or SGD. Rather, it shows that explicitly aiming for flat minima does not necessarily improve population risk performance (in the sense that establishing stronger risk upper bounds is impossible) and can sometimes even hurt it significantly. In addition, our analysis is limited to the convex setting and uses a strong notion of flatness. These assumptions actually make our lower bounds stronger, since they show that even in this simple setting and under a very strict definition of flatness, targeting flat solutions can still lead to worse performance.

**Open questions and future work.** Although our work is the first to discuss formally the limitations of sharpness-aware minimization in SCO, many questions remain open. Our lower-bounds constructions require a dimension that is exponential in the size of the training set, and reducing this dimensional dependence is an important open problem. Our result for SA-ERM shows that an arbitrary flat ERM solution may overfit, even though our construction also admits flat ERMs with good generalization. An interesting open question is whether one can construct instances in which all flat ERM solutions generalize poorly. Another direction for future work is to close the remaining gap between our upper and lower bounds. Further, our lower bounds for SA-GD and SAM focus on regimes where the perturbation radius of the algorithm exceeds the true flatness radius of the loss, understanding what happens when the perturbation radius is smaller than or comparable to the flatness radius remains open. Finally, it would be interesting to investigate whether similar phenomena arise for other variants of SAM.

## Acknowledgments

This project has received funding from the European Research Council (ERC) under the European Union's Horizon 2020 research and innovation program (grant agreement No. 101078075). Views and opinions expressed are however those of the author(s) only and do not necessarily reflect those of the European Union or the European Research Council. Neither the European Union nor the granting authority can be held responsible for them. This work received additional support from the Israel Science Foundation (ISF, grant numbers 2549/19 and 3174/23), a grant from the Tel Aviv University Center for AI and Data Science (TAD) and from the Len Blavatnik and the Blavatnik Family foundation.

In addition, this work was partially supported by the TAD Excellence Program for Doctoral Students in Artificial Intelligence and Data Science from the Tel Aviv University Center for AI and Data Science (TAD).

## Impact Statement

This paper presents work whose goal is to advance the field of Machine Learning. There are many potential societal consequences of our work, none which we feel must be specifically highlighted here.

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

# A. Proofs for Section 3

*Proof of Theorem 3.1.* Let $d = 2^n + 1$, $\mathcal{Z} = \{0, 1\}^{2^n}$ and let $\mathcal{D}$ be the uniform distribution over $\mathcal{Z}$. Consider the following function:

$$f(w, z) = \frac{1}{2} \max \left\{ \sqrt{\sum_{i=1}^{2^n} z(i) w(i)^2 + w(d)^2} - \rho, 0 \right\}^2.$$

We show that $f$ is convex, 1-Lipschitz, 1-smooth and has flatness radius $\rho$ in Lemma A.1. Since the samples are uniform over $\{0, 1\}^{2^n}$ we have that for a random training set $S = \{z_1, \ldots, z_n\} \overset{\text{i.i.d.}}{\sim} \mathcal{D}^n$ with probability greater than $1 - e^{-1} > \frac{1}{2}$, there exists an index $I \in [2^n]$ such that for every $z \in S$, $z(I) = 0$. From now on we will assume the existence of this $I$ and denote $w^{(1)} = e_I$. We will show $w^{(1)} \in \arg\min_{w \in W} F_S^r(w)$. First we notice that:

$$F_S^r(w, z) \geq F_S(w + \text{sign}(w(d)) \cdot r e_d) \geq \frac{1}{2} \max\{r - \rho, 0\}^2.$$

We will show that $w^{(1)}$ achieve that minimum. Indeed, from the choice of $I$, $F_S(e_I) = 0$, so for any $\|v\| \leq r$,

$$\begin{aligned} F_S(e_I + v) &= \frac{1}{2} \max \left\{ \|(e_I + v) \odot z\| - \rho, 0 \right\}^2 \\ &= \frac{1}{2} \max \left\{ \|v \odot z\| - \rho, 0 \right\}^2 \\ &\leq \frac{1}{2} \max \left\{ \|v\| - \rho, 0 \right\}^2 \leq \frac{1}{2} \max \left\{ r - \rho, 0 \right\}^2, \end{aligned}$$

which concludes the proof. Finally, since with probability $\frac{1}{2}$ a new sample $z'$ will hold $z'(I) = 1$:

$$F(e_I) - F(w^\star) \geq \frac{1}{4} \cdot (1 - \rho)^2 + \frac{1}{2} \cdot 0 \geq \frac{1}{16} = \Omega(1),$$

where the last inequality holds since $\rho \leq \frac{1}{2}$. This concludes the results for $w^{(1)}$. For $w^{(2)}$ consider $w^{(2)} = \rho e_d$. It is easy to see $w^{(2)}$ is a minimum of $F_S$ and for any $\delta > 0$:

$$F_S^\delta(\rho e_d) \geq F_S((\rho + \delta)e_d) = \frac{1}{2}(\rho + \delta - \rho)^2 = \frac{\delta^2}{2},$$

which shows $w^{(2)}$ is a sharp minimum. But,

$$F(\rho e_I) - F(w^\star) = \frac{1}{2} \max\{\rho - \rho\}^2 - 0 = 0.$$

which concludes the proof. $\square$

**Lemma A.1.** *Fix some $z \in [-1, 1]^{d-1} \times \{1\}$, and $\rho \geq 0$. Define the following function:*

$$\phi_z(w) = \frac{1}{2} \left[ \| [w \odot z]_+ \| - \rho \right]_+^2,$$

*then $\phi_z$ is convex, 1-Lipschitz, 1-smooth in the unit ball, and $\rho$-flat.*

*Proof of Lemma A.1.* We will prove each property separately.

**Convexity.** Notice that $\| [w \odot z]_+ \|$ is convex and the function $\psi(x) = \max\{x - \rho, 0\}^2$ is convex and non-decreasing, hence the composition $\phi_z = \psi \circ \xi_z$ is convex.

**Lipschitz continuity.** We will start by computing the gradient.

$$\begin{aligned} \|\nabla \phi_z(w)\| &= \left\| \frac{[\| [w \odot z]_+ \| - \rho, 0]_+}{\| [w \odot z]_+ \|} \cdot (z \odot [w \odot z]_+) \right\| \\ &\leq \|z \odot [w \odot z]_+\| \leq \|z\|_\infty \| [w \odot z]_+ \| \leq \|z\|_\infty \|w\| \\ &\leq \|w\| \leq 1, \end{aligned} \qquad (z \in [-1, 1]^d)$$

where the last inequality comes from the choice of the domain to be the unit ball.

**Smoothness.** For $x, y \in \mathbb{R}^d$,

$$\|\nabla\phi_z(x) - \nabla\phi_z(y)\| = \left\| z \odot \left( \frac{\left[\|x \odot z\| - \rho\right]_+}{\|x \odot z\|} [x \odot z]_+ - \frac{\left[\|y \odot z\| - \rho\right]_+}{\|y \odot z\|} [y \odot z]_+ \right) \right\|$$

$$\leq \|z\|_\infty \cdot \left\| \frac{\left[\|x \odot z\| - \rho\right]_+}{\|x \odot z\|} [x \odot z]_+ - \frac{\left[\|y \odot z\| - \rho\right]_+}{\|y \odot z\|} [y \odot z]_+ \right\|$$

$$\leq \left\| \frac{\left[\|x \odot z\| - \rho\right]_+}{\|x \odot z\|} [x \odot z]_+ - \frac{\left[\|y \odot z\| - \rho\right]_+}{\|y \odot z\|} [y \odot z]_+ \right\|. \qquad (z \in [-1, 1]^d)$$

Denote

$$T(u) := \frac{\left[\|u\| - \rho\right]_+}{\|u\|} u, \qquad (T(0) := 0),$$

so the last norm is $\|T([x \odot z]_+) - T([y \odot z]_+)\|$. Note the identity

$$T(u) = u - \Pi_{B_\rho}(u), \qquad B_\rho := \{v \in \mathbb{R}^d : \|v\| \leq \rho\},$$

where $\Pi_{B_\rho}$ is the Euclidean projection onto $B_\rho$. Also, using the fact that Euclidean projection is firmly nonexpansive, for any $u, v \in \mathbb{R}^d$:

$$\|T(u) - T(v)\|^2 = \| u - \Pi_{B_\rho}(u) - (v - \Pi_{B_\rho}(v)) \|^2$$

$$= \|u - v\|^2 + \|\Pi_{B_\rho}(u) - \Pi_{B_\rho}(v)\|^2 - 2\langle u - v, \Pi_{B_\rho}(u) - \Pi_{B_\rho}(v)\rangle$$

$$\leq \|u - v\|^2 + \|\Pi_{B_\rho}(u) - \Pi_{B_\rho}(v)\|^2 - 2\|\Pi_{B_\rho}(u) - \Pi_{B_\rho}(v)\|^2$$

$$= \|u - v\|^2 - \|\Pi_{B_\rho}(u - \Pi_{B_\rho}(v)\|^2$$

$$\leq \|u - v\|^2$$

Combining the inequalities we showed

$$\|\nabla\phi_z(x) - \nabla\phi_z(y)\| \leq \|[x \odot z]_+ - [y \odot z]_+\|$$

$$\leq \|x \odot z - y \odot z\| \qquad ([\cdot]_+ \text{ is 1-Lipschitz})$$

$$\leq \|z\|_\infty \|x - y\| \leq \|x - y\|. \qquad (z \in [-1, 1]^d)$$

This concludes the proof for smoothness.

**Flatness.** For $\rho$ flatness we can easily see that for any $\|v\| \leq \rho$ the following:

$$\phi_z(0 + v, \rho) = \phi_z(v, \rho) = \frac{1}{2} \left[ |[v \odot z]_+\| - \rho\right]_+^2 \leq \frac{1}{2} [\|v\| - \rho]_+^2 = 0$$

It is left to show that $\rho$ is the maximum flatness. Indeed, for every $w \in \arg\min \phi_z$:

$$\phi_z(w + \text{sign}(w(d)) \cdot c e_d) \geq \frac{1}{2} \max\{c - \rho, 0\}^2.$$

This implies that for $c > \rho$ we will have $\phi_z^c > 0$. $\qquad\qquad \square$

# B. Proofs for Section 4

### B.1. Proof of Theorem 4.1

In the proofs, we use the following standard lemma (e.g., (Srebro et al., 2010)).

**Lemma B.1.** *For a non-negative and $\beta$-smooth $f : \mathbb{R}^d \to \mathbb{R}$, it holds that $\|\nabla f(w)\|^2 \leq 2\beta f(w)$ for all $w \in \mathbb{R}^d$.*

*Proof of Lemma 4.2.* By Definition 2.1 we know that there exists a model $w^\star$ such that for every $\|v\| \le \rho$, it holds that $F_S(w^\star) = F_S(w^\star + v) = 0$. By Lemma B.1 and Young's inequality, since for every $t$, we know that $w_{t+1} = w_t - \eta \nabla F_S(w_t + v_t)$, it holds for every $\gamma > 0$ that,

$$\|w_{t+1} - w^\star\|^2 \le \|w_t - w^\star\|^2 - 2\langle \eta \nabla F_S(w_t + v_t), w_t - w^\star\rangle + \eta^2 \|\nabla F_S(w_t + v_t)\|^2$$

$$\le \|w_t - w^\star\|^2 - 2\left\langle \eta \nabla F_S(w_t + v_t), w_t + v_t - w^\star - \min\{\rho, \|v_t\|\}\frac{v_t}{\|v_t\|}\right\rangle$$

$$+ 2\left\langle \eta \nabla F_S(w_t + v_t), v_t - \min\{\rho, \|v_t\|\}\frac{v_t}{\|v_t\|}\right\rangle + 2\eta^2 \beta F_S(w_t + v_t) - 2\eta^2 \beta F_S(w^\star)$$

$$\le \|w_t - w^\star\|^2 - 2\eta F_S(w_t + v_t) + 2\eta F_S(w^\star) + \frac{1}{\gamma}\eta^2 \|\nabla F_S(w_t + v_t)\|^2 + \gamma \max\{r - \rho, 0\}^2$$

$$+ 2\eta^2 \beta F_S(w_t + v_t) - 2\eta^2 \beta F_S(w^\star).$$

For $\gamma = 4\eta\beta$ and $\eta \le \frac{1}{4\beta}$, we get that,

$$\|w_{t+1} - w^\star\|^2 \le \|w_t - w^\star\|^2 - 2\eta F_S(w_t + v_t) + 2\eta F_S(w^\star) + \frac{\eta}{4\beta}\|\nabla F_S(w_t + v_t)\|^2$$

$$+ 4\eta\beta \max\{r - \rho, 0\}^2 + 2\eta^2 \beta F_S(w_t + v_t) - 2\eta^2 \beta F_S(w^\star)$$

$$\le \|w_t - w^\star\|^2 - 2\eta F_S(w_t + v_t) + 2\eta F_S(w^\star) + \frac{\eta}{2}F_S(w_t + v_t) - \frac{\eta}{2}F_S(w^\star) +$$

$$4\eta\beta \max\{r - \rho, 0\}^2 + 2\eta^2 \beta F_S(w_t + v_t) - 2\eta^2 \beta F_S(w^\star) \qquad \text{(Lemma B.1)}$$

$$\le \|w_t - w^\star\|^2 + 4\eta\beta \max\{r - \rho, 0\}^2 - \eta F_S(w_t + v_t) + \eta F_S(w^\star)$$

Averaging from 1 to $T$ and rearragining, we get the lemma. $\qquad\square$

*Proof of Theorem 4.1.* Let $\bar{v} = \arg\max_{\|v\| \le r} F_S(v + \frac{1}{T}\sum_{t=1}^T w_t)$, thus, by Lemma 4.2, using Jensen inequality, we get

$$F_S^r\left(\frac{1}{T}\sum_{t=1}^T w_t\right) = F_S^r\left(\frac{1}{T}\sum_{t=1}^T w_t\right) - F_S(w^\star)$$

$$= F_S\left(\frac{1}{T}\sum_{t=1}^T w_t + \bar{v}\right) - F_S(w^\star)$$

$$\le \frac{1}{T}\sum_{i=1}^T F_S(w_t + \bar{v}) - F_S(w^\star)$$

$$\le \frac{1}{T}\sum_{i=1}^T F_S(w_t + v_t) - F_S(w^\star)$$

$$\le \frac{\|w_1 - w^\star\|^2}{\eta T} + 4\beta \max\{r - \rho, 0\}^2.$$

$\qquad\square$

## B.2. Proof of Theorem 4.3

*Proof of Theorem 4.3.* Let $\mathcal{Z} = \{-1, 1\}^{2^n}$ and $\mathcal{D}$ to be the uniform distribution over $\mathcal{Z}$. For $i \in [T]$ denote $w^{(i)} = w[T \cdot (i-1) + 1 : T \cdot i]$. Consider the following function:

$$f(w, z) = \frac{1}{2}\max\left\{\sqrt{\sum_{i=1}^{2^n}\sum_{j=1}^T \max\left\{z(i)w^{(i)}(j), 0\right\}^2 + w(d)^2} - \rho, 0\right\}^2.$$

we prove that $f$ is convex, 1-Lipschitz, 1-smooth and has flatness radius $\rho$ in Lemma A.1. From the definition of $\mathcal{D}$, for a sample $z \sim \mathcal{D}$ the coordinates $z(i)$ are i.i.d. uniform Bernoulli. For a random training set $S = \{z_1, \ldots, z_n\} \overset{\text{i.i.d.}}{\sim} \mathcal{D}^n$,

$S \subseteq \{0,1\}^{2^n}$, we have that with probability greater than $1 - e^{-1} > \frac{1}{2}$, there exists a coordinate $I$ such that all the examples in the sample are 1 on this coordinate, that is $z(I) = 1$ for all $z \in S$. Define the following SAM-gradient-oracle which at step $t$ outputs:

$$O_S^t(w) = \frac{1}{n} \sum_{i=1}^{n} \nabla f(w + e_{I_t}, z_i),$$

for $I_t = I + t - 1$. We will prove correctness by induction on $t$. For $w_1 = 0$ for every $\|v\| \leq r$ the following holds:

$$\frac{1}{n} \sum_{k=1}^{n} f(0 + v, z_k) = \frac{1}{2n} \sum_{k=1}^{n} \max \left\{ \sqrt{\sum_{i=1}^{2n} \sum_{j=1}^{T} \max \left\{ z_k(i)(0 + v^{(i)}(j)), 0 \right\}^2} - \rho, 0 \right\}^2$$

$$\leq \frac{1}{2n} \sum_{k=1}^{n} \max \left\{ \sqrt{\sum_{i=1}^{2n} \sum_{j=1}^{T} \max \left\{ v^{(i)}(j), 0 \right\}^2} - \rho, 0 \right\}^2$$

$$\leq \frac{1}{2n} \sum_{k=1}^{n} \max \left\{ \|v\| - \rho, 0 \right\}^2 \leq \frac{1}{2}(r - \rho)^2.$$

Also for $I_1$ chosen by the oracle:

$$\frac{1}{2n} \sum_{k=1}^{n} f(0 + e_I, z_k) = \frac{1}{2n} \sum_{k=1}^{n} \max \left\{ \sqrt{\sum_{i=1}^{2n} \sum_{j=1}^{T} \max \left\{ z_k(i)(0 + v^{(i)}(j)), 0 \right\}^2} - \rho, 0 \right\}^2$$

$$= \frac{1}{2n} \sum_{k=1}^{n} \max \left\{ \sqrt{\max \left\{ 0 + v(I), 0 \right\}^2} - \rho, 0 \right\}^2 = \frac{1}{2}(r - \rho)^2,$$

this concludes the base case. For the induction step we can notice that in step $t$ it holds that $w(i) \leq 0$ for every $i$ and $w(I_t) = 0$ thus the same steps as the base case complete the proof. To see no projections take place we note that by definition:

$$O_S^t(w_t) = \frac{1}{2n} \sum_{k=1}^{n} 2 \left( \sqrt{z_k(I_t)(w_t(I_t) + r)^2} - \rho \right) \cdot \frac{z \odot ([w + re_{I_t}]_+)}{\sqrt{z_k(I_t)(w_t(I_t) + r)^2}} = (r - \rho) \cdot \frac{re_{I_t}}{r} = (r - \rho)e_{I_t}.$$

This implies that at time $t$:

$$w_t(i) = \begin{cases} -\eta(r - \rho) & i \in \{I_j\}_{j=1}^{t-1} \\ 0 & \text{o.w.} \end{cases}.$$

Since $\eta(r - \rho) \leq \frac{1}{\sqrt{T}}$, we stay inside the unit ball for the entire run of the algorithm. This dynamics also imply that for every $\tau \in [T]$ suffix average $\widehat{w}_\tau = \frac{1}{T - \tau + 1} \sum_{t=\tau}^{T} w_t$ and $s \leq \frac{T}{2}$ the following holds:

$$\widehat{w}_\tau(I_s) = \frac{1}{T - \tau + 1} \sum_{t=\tau}^{T} w_t(I_s) \leq \frac{1}{T - \tau + 1} \sum_{t=\max\{\tau, T/2\}}^{T} w_t(I_s)$$

$$\leq \frac{T - \max\{\tau, T/2\} + 1}{T - \tau + 1} (-\eta(r - \rho)) \leq -\frac{\eta(r - \rho)}{2}.$$

With probability $\frac{1}{2}$ a new sample $z'$ will hold $z(I) = -1$ which gives:

$$F(\widehat{w}_\tau) - F(0) \geq \frac{1}{4} \max \left\{ \sqrt{\sum_{t=1}^{T} \widehat{w}_\tau^{(I)}(t)^2} - \rho, 0 \right\}^2 \geq \frac{1}{4} \max \left\{ \frac{\eta(r - \rho)}{2} \sqrt{\frac{T}{2}} - \rho, 0 \right\}^2$$

$$\geq \frac{1}{4} \max \left\{ \frac{\eta(r - \rho)}{2} \sqrt{\frac{T}{2}} - \frac{\eta(r - \rho)\sqrt{T}}{3}, 0 \right\}^2 \qquad (\rho \leq r - \frac{3r}{3 + \eta\sqrt{T}})$$

$$\geq \frac{1}{4 \cdot 100^2} \eta(r - \rho) \sqrt{T} = \Omega(\eta^2(r - \rho)^2 T).$$

$\square$

## B.3. Proof of Theorem 4.4

Our population loss upper bound for SA-GD (Theorem 4.4) are based on algorithmic stability (e.g., (Bousquet & Elisseeff, 2002; Hardt et al., 2016)). In this section, we revisit the main arguments required for these proofs and establish an algorithmic stability upper bound for first-order methods that minimize the SAER. In particular, the stability bounds in this section hold for any algorithm that produces a sequence $\{w_t\}_{t=1}^T$ satisfying $w_{t+1} = w_t - \eta \nabla F_S(w_t + v_t)$, where $\{v_t\}_{t=1}^T$ is a sequence of vectors such that for every $t$, $\|v_t\| \le r$ and $\eta \le 1/(2\beta)$.

The notion of stability that we consider is on-average-leave-one-out (loo) model stability (e.g., (Lei & Ying, 2020; Schliserman & Koren, 2022)). For this definition, we assume without loss of generality that there exists an example $z_0 \in \mathcal{Z}$ for which $f(w, z_0) = 0$ for all $w$. (Otherwise, we can artificially augment the sample space with such an instance.) Now, given an i.i.d. sample $S = (z_1, \ldots, z_n)$, with the corresponding $F_S$, we define the leave-one-out samples $S^{(i)} = (z_1, \ldots, z_{i-1}, z_0, z_{i+1}, \ldots, z_n)$ for all $i \in [n]$, with the corresponding empirical risks:

$$\forall i \in [n], \qquad F_{S^{(i)}} = \frac{1}{n} \sum_{z \in S_i} f(w, z) = \frac{1}{n} \sum_{j \ne i} f(w, z_j).$$

We can now define the on-average-loo model stability for learning algorithms.

**Definition B.2** ($\ell_2$-loo-on-average model stability). Let $A : \mathcal{Z}^n \to \mathbb{R}^d$ be a learning algorithm. We say that $A$ is $\ell_2$-on-average model $\epsilon$-stable if for any samples $S, S'$,

$$\frac{1}{n} \sum_{i=1}^n \|A(S) - A(S^{(i)})\|^2 \le \epsilon. \tag{6}$$

We will denote by $\epsilon_{\text{stab}}$ the infimum over all $\epsilon$ for which Eq. (6) holds.

Previous work has shown that an $\epsilon$-leave-one-out stable algorithm achieves good generalization. This is formalized in the following lemma from (Schliserman & Koren, 2022).

**Lemma B.3** (Lemma 7 from (Schliserman & Koren, 2022)). *Let $A$ be an $\ell_2$-on-average-loo model $\epsilon$-stable learning algorithm. Then, if for every $z$, $f(w, z)$ is convex and $\beta$-smooth with respect to $w$,*

$$\mathbb{E}F(A(S)) \le 4\mathbb{E}\left[F_S(A(S))\right] + 3\beta\epsilon.$$

We can now state the stability upper bound that we establish. It is formalized in the following lemma,

**Lemma B.4.** *Assume that for every $z$, $f(w, z)$ is $\beta$-smooth, convex, non-negative and $\rho$-flat. Let $A$ be an algorithm that given a data set $S$, produce a sequence $\{w_t\}_{t=1}^T$ such that*

$$w_{t+1} = w_t - \eta \nabla F_S(w_t + v_t),$$

*where $\{v_t\}_{t=1}^T$ are vectors such that for every $t$, $\|v_t\| \le r$ and $\eta \le 1/2\beta$. Assume that $A$ returns the averaged iterate $\widehat{w} := \frac{1}{T} \sum_{t=1}^T w_t$. Then, $A$ is $\ell_2$-on-average model $\epsilon$-stable with*

$$\epsilon_{\text{stab}} \le O\left(\eta\beta r^2 T + \frac{\beta\eta T}{n^2} + \frac{\beta^2\eta^2 T^2 \max(r - \rho, 0)^2}{n^2}\right)$$

The proof of Lemma B.4 appears in Appendix B.3.1. We will now prove Theorem 4.4.

*Proof of Theorem 4.4.* By Theorem 4.1, we know that

$$F_S\left(\frac{1}{T} \sum_{t=1}^T w_t\right) \le F_S^r\left(\frac{1}{T} \sum_{t=1}^T w_t\right) \le \frac{\|w_1 - w^\star\|^2}{\eta T} + 4\beta \max\{r - \rho, 0\}^2.$$

By Lemma B.4, we know that, the algorithm is $\ell_2$-on-average model $r$-stable with

$$\epsilon \le 24\eta\beta r^2 T + \frac{96\beta\eta T}{n^2} + \frac{768\beta^2\eta^2 T^2 \max(r - \rho, 0)^2}{n^2}.$$

By combining both equations with Lemma B.3 we get the theorem. $\qquad\square$

### B.3.1. OMITTED PROOFS

*Proof of Lemma B.4.* Denote by $\{w_t^{(i)}\}_{t \in [T]}$ the iterates of $S^{(i)}$ and by $\{v_t^{(i)}\}_{t \in [T]}$ the corresponding sequence of perturbations vectors. It holds that,

$$
\begin{aligned}
\|w_{t+1} - w_{t+1}^{(i)}\|^2 &= \|w_t - w_t^{(i)} - \eta(\nabla F_S(w_t + v_t) - \nabla F_{S^{(i)}}(w_t^{(i)} + v_t^{(i)}))\|^2 \\
&\le \|w_t - w_t^{(i)}\|^2 + \underbrace{\eta^2 \|\nabla F_S(w_t + v_t) - \nabla F_{S^{(i)}}(w_t^{(i)} + v_t^{(i)})\|^2}_{(I)} \\
&\quad - \underbrace{2\eta \langle \nabla F_S(w_t + v_t) - \nabla F_{S^{(i)}}(w_t^{(i)} + v_t^{(i)}), w_t - w_t^{(i)} \rangle}_{(II)}
\end{aligned}
$$

Treating the two terms (I),(II) separately, for (I) it holds by Lemma B.1 that,

$$
\begin{aligned}
&\eta^2 \|\nabla F_S(w_t + v_t) - \nabla F_{S^{(i)}}(w_t^{(i)} + v_t^{(i)})\|^2 \\
&\le 2\eta^2 \|\nabla F_{S^{(i)}}(w_t + v_t) - \nabla F_{S^{(i)}}(w_t^{(i)} + v_t^{(i)})\|^2 + \frac{2\eta^2}{n^2} \|\nabla f(w_t + v_t, z_i)\|^2 \\
&\le 2\eta^2 \|\nabla F_{S^{(i)}}(w_t + v_t) - \nabla F_{S^{(i)}}(w_t^{(i)} + v_t^{(i)})\|^2 + \frac{4\eta^2}{n^2} \|\nabla f(w_t + v_t, z_i\|^2 \\
&\le 2\eta^2 \|\nabla F_{S^{(i)}}(w_t + v_t) - \nabla F_{S^{(i)}}(w_t^{(i)} + v_t^{(i)})\|^2 + \frac{8\beta\eta^2}{n^2} f(w_t + v_t, z_i).
\end{aligned}
$$

For (II), it holds by two uses of Young's inequality that,

$$
\begin{aligned}
&- 2\eta \langle \nabla F_S(w_t + v_t) - \nabla F_{S^{(i)}}(w_t^{(i)} + v_t^{(i)}), w_t - w_t^{(i)} \rangle \\
&= - 2\eta \langle \nabla F_{S^{(i)}}(w_t + v_t) - \nabla F_{S^{(i)}}(w_t^{(i)} + v_t^{(i)}), w_t - w_t^{(i)} \rangle - \frac{2\eta}{n} \langle \nabla f(w_t + v_t, z_i), w_t - w_t^{(i)} \rangle \\
&= - 2\eta \langle \nabla F_{S^{(i)}}(w_t + v_t) - \nabla F_{S^{(i)}}(w_t^{(i)} + v_t^{(i)}), w_t + v_t - w_t^{(i)} - v_t^{(i)} \rangle \\
&\quad - \frac{2\eta}{n} \langle \nabla f(w_t + v_t, z_i), w_t - w_t^{(i)} \rangle + 2\eta \langle \nabla F_{S^{(i)}}(w_t + v_t) - \nabla F_{S^{(i)}}(w_t^{(i)} + v_t^{(i)}), v_t - v_t^{(i)} \rangle \\
&\le - \frac{2\eta}{\beta} \|\nabla F_{S^{(i)}}(w_t + v_t) - \nabla F_{S^{(i)}}(w_t^{(i)} + v_t^{(i)})\|^2 \\
&\quad + \frac{\eta}{\alpha n} \|w_t - w_t^{(i)}\|^2 + \frac{\eta\alpha}{n} \|\nabla f(w_t + v_t, z_i)\|^2 \\
&\quad + \frac{\eta}{\gamma} \|\nabla F_{S^{(i)}}(w_t + v_t) - \nabla F_{S^{(i)}}(w_t^{(i)} + v_t^{(i)})\| + \eta\gamma \|v_t - v_t^{(i)}\|^2
\end{aligned}
$$

By setting $\alpha = \eta T/n$ and using co-coercivity of-gradients of smooth functions, we get,

$$
\begin{aligned}
&- 2\eta \langle \nabla F_S(w_t + v_t) - \nabla F_{S^{(i)}}(w_t^{(i)} + v_t^{(i)}), w_t - w_t^{(i)} \rangle \\
&\le (\frac{\eta}{\gamma} - \frac{2\eta}{\beta}) \|\nabla F_{S^{(i)}}(w_t + v_t) - \nabla F_{S^{(i)}}(w_t^{(i)} + v_t^{(i)})\|^2 + \frac{\eta}{\alpha n} \|w_t - w_t^{(i)}\|^2 \\
&\qquad\qquad\qquad\qquad\qquad\qquad\qquad + \frac{2\beta\alpha\eta}{n} f(w_t + v_t, z_i) + 4\eta\gamma r^2 \\
&\le (\frac{\eta}{\gamma} - \frac{2\eta}{\beta}) \|\nabla F_{S^{(i)}}(w_t + v_t) - \nabla F_{S^{(i)}}(w_t^{(i)} + v_t^{(i)})\|^2 + \frac{1}{T} \|w_t - w_t^{(i)}\|^2 \\
&\qquad\qquad\qquad\qquad\qquad\qquad\qquad + \frac{2\beta\eta^2 T}{n^2} f(w_t + v_t, z_i) + 4\eta\gamma r^2.
\end{aligned}
$$

Averaging over $i \in [n]$, plugging both in, and setting $\gamma = \beta, \eta \leq \frac{1}{2\beta}$

$$\frac{1}{n}\sum_{i=1}^{n}\|w_{t+1} - w_{t+1}^{(i)}\|^2$$

$$\leq \left(1 + \frac{1}{T}\right)\frac{1}{n}\sum_{i=1}^{n}\|w_t - w_t^{(i)}\|^2 + \frac{8\beta\eta^2(T+1)}{n^2}F_S(w_t + v_t)$$

$$+ 4\eta\gamma r^2 + (2\eta^2 - \frac{2\eta}{\beta} + \frac{\eta}{\gamma})\|\nabla F_{S^{(i)}}(w_t + v_t) - \nabla F_{S^{(i)}}(w_t^{(i)} + v_t^{(i)})\|^2$$

$$\leq \left(1 + \frac{1}{T}\right)\frac{1}{n}\sum_{i=1}^{n}\|w_t - w_t^{(i)}\|^2 + \frac{8\beta\eta^2(T+1)}{n^2}F_S(w_t + v_t) + 4\eta\beta r^2$$

$$\leq \frac{e^{\frac{1}{T}}}{n}\sum_{i=1}^{n}\|w_t - w_t^{(i)}\|^2 + \frac{8\beta\eta^2(T+1)}{n^2}F_S(w_t + v_t) + 4\eta\beta r^2.$$

Now, unrolling the recursion, we get,

$$\frac{1}{n}\sum_{i=1}^{n}\|w_{t+1} - w_{t+1}^{(i)}\|^2 \leq \sum_{t=1}^{T}e^{\frac{T-t}{T}}\left(\frac{8\beta\eta^2(T+1)}{n^2}F_S(w_t + v_t) + 4\eta\beta r^2\right)$$

$$\leq \frac{24\beta\eta^2(T+1)}{n^2}\sum_{t=1}^{T}F_S(w_t + v_t) + 12\eta\beta r^2 T.$$

Using Lemma 4.2, we get for every $t$ that,

$$\frac{1}{n}\sum_{i=1}^{n}\|w_{t+1} - w_{t+1}^{(i)}\|^2 \leq \frac{96\beta\eta^2 T}{n^2}\left(\frac{1}{\eta} + 4\beta T\max(r - \rho, 0)^2\right) + 12\eta\beta r^2 T$$

$$= 12\eta\beta r^2 T + \frac{96\beta\eta T}{n^2} + \frac{384\beta^2\eta^2 T^2\max(r - \rho, 0)^2}{n^2}.$$

By Jensen's inequality and the convexity of squared $\ell_2$ norm, we get that,

$$\frac{1}{n}\sum_{i=1}^{n}\|\frac{1}{T}\sum_{i=1}^{T}w_t - \frac{1}{T}\sum_{i=1}^{T}w_t^{(i)}\|^2 \leq 24\eta\beta r^2 T + \frac{96\beta\eta T}{n^2} + \frac{768\beta^2\eta^2 T^2\max(r - \rho, 0)^2}{n^2}.$$

$\square$

## C. Proofs for Section 5

### C.1. Proof of Theorem 5.1

*Proof of Theorem 5.1.* By the convexity of $F_S$ we know that, for every $t$,

$$F_S(w_t + v_t) \geq F_S(w_t) + \langle\nabla F_S(w_t), v_t\rangle$$

$$= F_S(w_t) + \langle\nabla F_S(w_t), r\frac{\nabla F_S(w_t)}{\|\nabla F_S(w_t)\|}\rangle$$

$$= F_S(w_t) + r\|\nabla F_S(w_t)\|$$

$$\geq F_S(w_t).$$

Then, by Lemma 4.2, using Jensen inequality, we get,

$$F_S \left( \frac{1}{T} \sum_{t=1}^{T} w_t \right) \leq \frac{1}{T} \sum_{i=1}^{T} F_S(w_t) - F_S(w^\star)$$

$$\leq \frac{1}{T} \sum_{i=1}^{T} F_S(w_t + v_t) - F_S(w^\star)$$

$$\leq \frac{\|w_1 - w^\star\|^2}{\eta T} + 4\beta \max\{r - \rho, 0\}^2.$$

$\square$

### C.2. Proof of Theorem 5.2

*Proof of Theorem 5.2.* Let $f(w) = \frac{1}{2} \max(0, x)^2$. Its (one-dimensional) derivatives are, for $w \neq 0$,

$$f'(w) = w, \qquad f''(w) = 1,$$

and for $w < 0$,

$$f'(w) = f''(w) = 0,$$

$f$ is a non-negative function. The convexity is implied by the positivity of $f''$. The Lipschitzness is implied by the fact that $|f'(w)| \leq 1$ for every $w \in W$. The smoothness is followed by the fact that $g(w) = \max(0, w)$ is a Lipschitz function as a max function over two Lipschitz functions. In addition, $f$ is $\rho$-flat since $w^\star = -\frac{1}{2}$, holds $f(w^\star + v) = 0$ for every $\|v\| \leq \frac{1}{2}$. Now, let $w_1 = 0$. Since $f'(0) = 0$, $w_2 = w_1 = 0$ and by induction it follows that SAM satisfies $w_t = 0$ for every $t$. As a result, for any $\tau$, $\widehat{w}_\tau = 0$, and, for every $0 \leq r \leq \frac{1}{2}$, it holds that,

$$F_S^r(\widehat{w}_\tau) - F_S^r(w^\star) = \max_{v \leq r} \frac{1}{2} \max(0, v)^2 - 0 = \frac{1}{2} r^2.$$

$\square$

### C.3. Proof of Theorem 5.3

*Proof of Theorem 5.3.* Let $d = T \cdot 2^n + 1$, $\mathcal{Z} = \{0, 1\}^{2^n}$, $\mathcal{D}$ to be the uniform distribution over $\mathcal{Z}$. Denote for every $i \in [T]$; $w^{(i)} = w[T \cdot (i-1) + 1 : T \cdot i]$. Consider the following function:

$$f(w, z) = \frac{1}{2} \sum_{i=1}^{2^n} \sum_{j=2}^{T} z(i) w^{(i)}(j)^2$$

$$+ \frac{1}{2} \sum_{i=1}^{2^n} \sum_{j=2}^{T} \max \left\{ w^{(i)}(j) - \delta_j \left( w^{(i)}(j-1) + \lambda \cdot \mathbb{1}[j = 2] \right), 0 \right\}^2$$

$$+ \frac{\gamma}{2} \max\{v_z^T w + \delta_1, 0\}^2,$$

where

$$v_z^{(i)}(j) = \begin{cases} 0 & j \neq 1 \\ -\frac{1}{2(d-1)} & i \leq 2^n, \ j = 1 \text{ and } z(i) = 0 \\ 1 & i \leq 2^n, \ j = 1 \text{ and } z(i) = 1 \\ 1 & i = 2^n + 1 \text{ and } j = 1 \end{cases},$$

and,

$$\delta_1 = \frac{\eta \gamma r}{2\sqrt{d} - \eta \gamma}, \qquad \lambda = \frac{r}{4d(d-1)}, \qquad \gamma = \frac{\lambda}{\max\{1, \eta\}(r + \delta_1)}.$$

The positive parameters $\{0 < \delta_j \leq 1\}_{j=2}^T$ will be chosen later. We will prove $f$ has the desired properties in the following lemma whose proof is deferred to Appendix C.3.1.

**Lemma C.1.** *$f$ defined as defined above is convex, 6-smooth, 7-Lipschitz and realizable, meaning $\rho$-flat with $\rho = 0$.*

Since the distribution $\mathcal{D}$ is uniform over $\{0, 1\}^{2^n}$, for a random training set $S = \{z_1, \ldots, z_n\}$ with probability at least $\frac{1}{e} > \frac{1}{3}$, there exists *exactly one* index $I$ such that for every $z \in S$, $z(I) = 0$. For the rest of the proof, assume this event holds. We will show the dynamics of the algorithm under this assumption in the following lemma which proof is deferred to Appendix C.3.1:

**Lemma C.2.** *Assuming there exists a coordinate $I$ such that $\forall z \in S$; $z(I) = 0$, and $\eta r \leq \frac{1}{\sqrt{T}}$, $w_1 = 0$, then there exists $\delta_2 > 0$ such that after running one SAM update on $F_S$,*

1. $\forall z \in S$; $v_z^T w_2 + \delta_1 \leq 0$

2. $\forall i \neq I$; $-\lambda < w_2^{(i)}(1) < 0$

3. $\forall i \neq I$, $j \geq 2$; $w_2^{(i)}(j) = 0$

4. $\forall j \geq 3$; $w_2^{(I)}(j) = 0$

5. $0 \leq w_2^{(I)}(1) \leq \frac{1}{d}$

6. $-\frac{1}{d} \leq w_2^{(I)}(2) < 0$.

From this lemma we can conclude that if $w_t^{(I)}(2)$ remains negative throughout the remaining run of the algorithm, none of the coordinates in $w^{(i)}$ where $i \neq I$ will change, and neither will $w^{(i)}(1)$ for every $i$. This means that while $w_t^{(I)}(2)$ remains negative it suffices to prove the dynamics for the following function:

$$g(u) = \frac{1}{2} \sum_{j=3}^{T} \max \left\{ u(j) - \delta_j u(j-1), 0 \right\}^2 + \max\{u(2), 0\}^2,$$

when we start from $u_2 = -\sigma e_2$ for $\sigma = |w_2^{(I)}(2)| > 0$. The dynamics we will prove for $u[2 : T]$ will hold for $w^{(I)}[2 : T]$ while the rest of $w$ stays the same as in $w_2$. We will now continue to look at the dynamics of $\{u_t\}_{t=2}^{T}$. We will have the following lemma whose proof is deferred to Appendix C.3.1:

**Lemma C.3.** *There exists a set of positive parameters $\{0 < \delta_t \leq 1\}_{t=3}^{T}$ such that starting from $u_2 = -\sigma e_2$ will give us the following for $t \geq 4$:*

1. $-\sigma \leq u_t(2) \leq 0$

2. $u_t(i+1) - \delta_i u_t(i) \begin{cases} \leq 0 & 2 \leq i < t \\ > 0 & i = t \\ = 0 & t < i \leq T - 1 \end{cases}$

3. $-2\eta r \leq u_t(t) \leq -\eta r$

4. $-2\eta r \leq u_t(t-1) \leq -\frac{1}{2}\eta r$.

In the proof of the dynamic of $u$ we did not consider projections, that is because with this dynamic and the assumption that $\eta r \leq \frac{1}{2\sqrt{T}}$ means we stay inside the unit ball for the entire algorithm and no projections take place. To see this notice using Lemmas C.2 and C.3 that for every $t \in [T]$:

$$\|w_t\|^2 \leq \|w_T\|^2 \leq 2(T-1) \cdot 4\eta^2 r^2 + d \cdot \frac{1}{d^2} \leq 4\frac{(T-1)}{4T} + \frac{1}{T \cdot 2^n} \leq 1.$$

Concluding we know that for $t = 3, \ldots, T$:

$$\forall j \in \{3, \ldots, t\}; \ w_t^{(I)}(j) \leq -\frac{1}{2}\eta r.$$

This implies that for a suffix average $\tau \in [T]$; $\widehat{w}_\tau = \frac{1}{T-\tau+1} \sum_{t=\tau}^T w_t$ we have that for $s \geq \frac{T}{2}$:

$$\widehat{w}_\tau^{(I)}(s) = \frac{1}{T-\tau+1} \sum_{t=\tau}^T w_t^{(I)}(s) \leq \frac{1}{T-\tau+1} \sum_{t=\max\{\tau, T/2\}}^T w_t^{(I)}(s)$$

$$\leq \frac{T - \max\{\tau, T/2\} + 1}{T - \tau + 1} \left( -\frac{1}{2}\eta r \right) \leq -\frac{\eta r}{4}.$$

With probability $\frac{1}{2}$ a new sample $z'$ will have $z'(I) = 1$. This means that for every $\tau \in [T]$:

$$F(\widehat{w}_\tau) - F(w^\star) \geq \|\widehat{w}_\tau\|^2 \geq \frac{1}{4}\eta^2 r^2 \cdot \frac{T}{2} = \Omega(\eta^2 r^2 T).$$

Where we use the fact that $F$ is realizable. This concludes the proof. $\qquad\square$

### C.3.1. OMITTED PROOFS

*Proof of Lemma C.1.* We will use the following notation:

$$f(w, z) = \underbrace{\frac{1}{2} \sum_{i=1}^{2^n} \sum_{j=2}^T z(i)\, w^{(i)}(j)^2}_{=:f_1(w)}$$

$$+ \underbrace{\frac{1}{2} \sum_{i=1}^{2^n} \sum_{j=2}^T \left[ w^{(i)}(j) - \delta_j\big(w^{(i)}(j-1) + \lambda \mathbb{1}[j=2]\big) \right]_+^2}_{=:f_2(w)}$$

$$+ \underbrace{\frac{\gamma}{2} \left[ v_z^\top w + \delta_1 \right]_+^2}_{=:f_3(w)},$$

**Convexity.** Each component is convex:

- $f_1$: a nonnegative sum of convex quadratics.

- $f_2$: each term is $\frac{1}{2}(\text{affine}(w))_+^2$, convex because $x \mapsto \frac{1}{2}(x_+)^2$ is convex and nondecreasing.

- $f_3$: same reasoning as $f_2$.

Therefore $f$ is convex.

**Lipschitz continuity.** We will bound the norm of the gradients inside the unite ball.

- $f_1$: $\nabla f_1(w) = z(i)\, w^{(i)}(j)$ on each $(i, j)$ with $j \geq 2$, hence $\|\nabla f_1(w)\| \leq \|z\|_\infty \|w\| \leq 1$.

- $f_2$: define $r_{i,j}(w) = \left[ w^{(i)}(j) - \delta_j(w^{(i)}(j-1) + \lambda\mathbb{1}[j=2]) \right]$. Each term $\frac{1}{2}[r_{i,j}(w)]_+^2$ contributes gradient supported on $w^{(i)}(j), w^{(i)}(j-1)$ with squared norm $(1 + \delta_j^2)[r_{i,j}(w)]_+^2 \leq 2[r_{i,j}(w)]_+^2 \leq 2r_{i,j}(w)^2$. Using the fact that

$(a - b)^2 \leq 2a^2 + 2b^2$ for any $a, b$, we get:

$$
\|\nabla f_2(w)\|^2 = \left\| \sum_{i=1}^{2^n} \sum_{j=2}^{T} \nabla \left( \frac{1}{2} [r_{i,j}(w)]_+^2 \right) \right\|^2 \leq \sum_{i=1}^{2^n} \sum_{j=2}^{T} \left\| \nabla \left( \frac{1}{2} [r_{i,j}(w)]_+^2 \right) \right\|^2 \leq \sum_{i=1}^{2^n} \sum_{j=2}^{T} 2 r_{i,j}(w)^2
$$

$$
= 2 \sum_{i=1}^{2^n} \sum_{j=2}^{T} \left( w^{(i)}(j) + \delta_j (w^{(i)}(j-1) + \lambda \mathbb{1}[j=2]) \right)^2
$$

$$
\leq 4 \sum_{i=1}^{2^n} \sum_{j=2}^{T} \left( w^{(i)}(j)^2 + \delta_j^2 (w^{(i)}(j-1)^2 + \lambda \mathbb{1}[j=2])^2 \right)
$$

$$
\leq 4 \sum_{i=1}^{2^n} \sum_{j=2}^{T} \left( w^{(i)}(j)^2 + 2\delta_j^2 w^{(i)}(j-1)^2 + 2\delta_j^2 \lambda \mathbb{1}[j=2])^2 \right)
$$

$$
\leq 12 \|w\|^2 + 8\delta_2^2 \lambda \cdot 2^n
$$

$$
\leq 12 + 8 \cdot 1 \cdot \frac{r}{4d(d-1)} \cdot 2^n \leq 13.
$$

Hence $f_2$ is 4-Lipschitz

- $f_3$: $\nabla f_3(w) = \gamma (v_z^\top w + \delta_1)_+ v_z$, hence $\|\nabla f_3(w)\|_2 \leq \gamma (\|v_z\| + |\delta_1|) \|v_z\| \leq \frac{1}{4d(d-1)} \cdot (\frac{d}{T} + 1) \cdot \frac{d}{T} \leq 1$.

Adding the three bounds gives that $f$ is 6-Lipschitz.

**Smoothness.**

- $f_1$: this function's Hessian is diagonal with entries $z(i)$ on coordinates $(i, j)$ with $j \geq 2$. Since $z(i) \in \{0, 1\}$, $f_1$ is 1-smooth.

- $f_2$: For each $i$, stack the variables as $w^{(i)} \in \mathbb{R}^T$ and define the linear map

$$
(Bw^{(i)})_{j-1} = w^{(i)}(j) - \delta_j w^{(i)}(j-1), \qquad j = 2, \ldots, T,
$$

so $B \in \mathbb{R}^{(T-1) \times T}$ has 1 on the superdiagonal and $-\delta_j$ on the subdiagonal positions that touch it. Let $b \in \mathbb{R}^{T-1}$ encode the constant shift $b_1 = -\delta_2 \lambda$ and $b_k = 0$ for $k \geq 2$. Writing $x$ for the full vector that stacks all $w^{(i)}$, we can express

$$
f_2(x) = \frac{1}{2} \sum_{i=1}^{2^n} \left\| [Bw^{(i)} + b]_+ \right\|_2^2 = \frac{1}{2} \left\| [Ax + c]_+ \right\|_2^2,
$$

where $A$ is block-diagonal with $2^n$ copies of $B$ and $c$ stacks the copies of $b$. By the chain rule,

$$
\nabla f_2(x) = A^\top [Ax + c]_+.
$$

Hence, for any $x, y$,

$$
\begin{aligned}
\|\nabla f_2(x) - \nabla f_2(y)\| &= \left\| A^\top ([Ax + c]_+ - [Ay + c]_+) \right\| \\
&\leq \|A\| \, \| [Ax + c]_+ - [Ay + c]_+ \| \\
&\leq \|A\| \, \|A(x - y)\| \leq \|A\|^2 \, \|x - y\|.
\end{aligned}
\qquad ([\cdot]_+ \text{ is 1-Lipschitz})
$$

Therefore $f_2$ is $\|A\|^2 = \|B\|^2$-smooth. Using $\delta_j \leq 1$ and $(a - b)^2 \leq 2a^2 + 2b^2$, for any $\|x\| = 1$:

$$
\|Bx\|_2^2 = \sum_{j=2}^{T} (x_j - \delta_j x_{j-1})^2 \leq 2 \sum_{j=2}^{T} x_j^2 + 2 \sum_{j=2}^{T} \delta_j^2 x_{j-1}^2 \leq 4 \sum_{j=1}^{T} x_j^2 \leq 4,
$$

so $\|B\|^2 \leq 4$ and consequently $f_2$ is 4-smooth.

- $f_3$: $\nabla f_3(w) = \gamma\,[\,v_z^\top w + \delta_1\,]_+\,v_z$. For any $x, y$,

$$\|\nabla f_3(x) - \nabla f_3(y)\| = \gamma\,\|[\,v_z^\top x + \delta_1\,]_+ - [\,v_z^\top y + \delta_1\,]_+\|\,\|v_z\| \le \gamma\,\|v_z^\top(x-y)\|\,\|v_z\|$$

$$\le \gamma\|v_z\|^2\|x-y\| \le \frac{d}{4d(d-1)}\|x-y\| \le \|x-y\|,$$

so $f_3$ is 1-smooth.

Summing gives $f$ is 6 smooth.

**Realizability.**  We can see that for

$$w^\star(i) = \begin{cases} 0 & i < d \\ -\frac{\lambda}{2} & i = d \end{cases}$$

$f(w^\star, z) = 0$ for every $z \in \{0, 1\}^d$.  $\qquad\square$

*Proof of Lemma C.2.*  Denote $v_S = \frac{1}{n}\sum_{k=1}^n v_{z_k}$. We will compute the gradient steps explicitly,

$$\nabla F(w_1) = \frac{1}{n}\sum_{k=1}^n \delta_1 \cdot \gamma v_{z_k} = \delta_1 \gamma v_S.$$

Hence,

$$w_{1+1/2} = 0 + \frac{r\delta_1\gamma v_S}{\delta_1\gamma\|v_S\|} = \frac{r v_S}{\|v_S\|}.$$

Since $n \ge 2$ it holds that $\frac{1}{2(d-1)} \le \frac{1}{2n}$. This implies $v_z(i) = 1 \implies w_{1+1/2}(i) > 0$. Hence, for every $z \in S$:

$$v_z^T w_{1+1/2} \ge w_1^{(2^n+1)}(1) - \frac{r}{2(d-1)} \cdot \frac{d-1}{\|v_S\|} = \frac{r}{2\|v_S\|} > 0.$$

We can calculate the first SAM update explicitly,

$$\nabla F_S(w_{1+1/2}) = \frac{1}{n}\sum_{k=1}^n \gamma\left(v_{z_k} \odot \frac{r v_S}{\|v_S\|} + \delta_1\right) \odot v_{z_k} + \left[-\delta_2\left(\frac{r v_S(I)}{\|v_S\|} + \lambda\right)\right]_+ (e_2^{(I)} - \delta_2 e_1^{(I)})$$

$$= \gamma\left(\frac{r v_S}{\|v_S\|}\left[\frac{1}{n}\sum_{k=1}^n v_{z_k} \odot v_{z_k}\right] + \delta_1 v_S\right) - \delta_2\left(\frac{r v_S(I)}{\|v_S\|} + \lambda\right)(e_2^{(I)} - \delta_2 e_1^{(I)}),$$

where the last step is from the fact that:

$$\frac{v_S(I)}{\|v_S\|} + \lambda = -\frac{r}{2(d-1)\|v_S\|} + \lambda \le -\frac{1}{2(d-1)d} + \frac{r}{4d(d-1)} = -\frac{r}{4d(d-1)} < 0.$$

Notice that similarly to before, this gradient step guarantees $v_z(i) = 1 \implies w_2(i) < 0$. Since $v_S(T \cdot 2^n + 1) = 1$, for every $z \in S$:

$$v_z^T w_2 \le w_2^{(2^n+1)}(1)\left(1 - \frac{1}{2(d-1)}(d-1)\right) = \frac{1}{2}w_2^{(2^n+1)}(1) = -\frac{\eta\gamma}{2}\left(\frac{r+\delta_1}{\|v_S\|}\right)$$

$$\le -\frac{\eta\gamma(r+\delta_1)}{2\sqrt{d}} < 0 \qquad\qquad (\|v_S\| \le \sqrt{d})$$

This implies that for every $z \in S$:

$$v_z^T w_2 + \delta_1 \le -\frac{\eta\gamma(r+\delta_1)}{2\sqrt{d}} + \delta_1 = \frac{-\eta\gamma r + \delta_1(2\sqrt{d} - \eta\gamma)}{2\sqrt{d}} = 0.$$

Where the last step is due to the choice of $\delta_1$ and concludes Item 1. Furthermore, for every $i \neq I$ we have that:

$$w_2^{(i)}(1) + \lambda \geq -\frac{\gamma(r + \delta_1)}{\|v_S\|} + \lambda \geq -\gamma(r + \delta_1) + \lambda = 0,$$

where the last step is from the choice of $\gamma$ concluding Item 2. Finally,

$$w_2^{(I)}(1) = -\eta\delta_2^2 \left(\frac{rv_S(I)}{\|v_S\|}\right) + \eta\gamma \left(\frac{r}{8(d-1)^3\|v_S\|} + \frac{\delta_1}{2(d-1)}\right)$$

$$\leq -\eta\delta_2^2 \left(\frac{rv_S(I)}{\|v_S\|}\right) + \frac{1}{4d(d-1)} \left(\frac{1}{8(d-1)^3} + \frac{1}{2(d-1)}\right),$$

where the last inequality is again from the choice of $\gamma$. This implies that there exists $\tau_1 > 0$ such that for every $\delta_2 \leq \tau_1$ it holds that $0 \leq w_2^{(I)}(1) \leq \frac{1}{\sqrt{d}}$. Similarly,

$$w_2^{(I)}(2) = \eta\delta_2 \left(\frac{rv_S(I)}{\|v_S\|} + \lambda\right).$$

Since this goes to 0 when $\delta_2$ goes to zero, there exists $\tau_2$ such that for every $0 < \delta_2 \leq \tau_2$; $-\frac{1}{\sqrt{d}} \leq w_2^{(I)}(2) < 0$. Choosing $0 < \delta_2 = \min\{\tau_1, \tau_2, 1\}$ concludes Items 5 and 6. Items 3 and 4 hold since these coordinates weren't changed by the update and thus stayed 0. □

*Proof of Lemma C.3.* We will show the claim by induction on $t$.

**Base case.** We will start by computing $u_4$. Using all we've proved we get:

$$\nabla g(u_2) = -\delta_3\sigma(e_3 - \delta_3 e_2),$$

which gives:

$$u_{2+1/2} = u_2 + \frac{r\nabla g(u_2)}{\|\nabla g(u_2)\|} = u_2 + \frac{r\delta_3}{\delta_3\sqrt{1+\delta_3^2}}e_3 - \frac{r\delta_3^2}{\delta_3\sqrt{1+\delta_3^2}}e_2$$

$$= u_2 + \frac{r}{\sqrt{1+\delta_3^2}}e_3 - \frac{r\delta_3}{\sqrt{1+\delta_3^2}}e_2.$$

Thus,

$$\nabla g(u_{2+1/2}) = (u_{2+1/2}(3) - \delta_3 u_{2+1/2}(2))e_3 - \delta_3(u_{2+1/2}(3) - \delta_3 u_{2+1/2}(2))e_2$$

$$= \left(\frac{r}{\sqrt{1+\delta_3^2}} + \frac{r\delta_3^2}{\sqrt{1+\delta_3^2}} - \delta_3 u_2(2)\right)e_3 - \delta_3\left(\frac{r}{\sqrt{1+\delta_3^2}} + \frac{r\delta_3^2}{\sqrt{1+\delta_3^2}} - \delta_3 u_2(2)\right)e_2$$

$$= \left(r\sqrt{1+\delta_3^2} - \delta_3 u_2(2)\right)e_3 - \delta_3\left(r\sqrt{1+\delta_3^2} - \delta_3 u_2(2)\right)e_2.$$

Finally,

$$u_3 = u_2 - \eta\left(r\sqrt{1+\delta_3^2} - \delta_3 u_2(2)\right)e_3 + \eta\delta_3\left(r\sqrt{1+\delta_3^2} - \delta_3 u_2(2)\right)e_2.$$

This gives:

$$-\sigma \leq u_3(2) = -\sigma + \eta\delta_3\left(r\sqrt{1+\delta_3^2} - \delta_3 u_2(2)\right).$$

Importantly $\sigma$ does not depend on $\delta_3$ so this term goes to $-\sigma < 0$ as $\delta_3$ goes to 0. This means that there exists $\tau_1$ such that for every $\delta_3 \leq \tau_1$ we have that $u_3(2) < 0$. Furthermore,

$$u_3(3) = -\eta \left( r\sqrt{1 + \delta_3^2} - \delta_3 u_2(2) \right) \leq -\eta r + \eta \delta_3 u_2(2) \leq -\eta r,$$

where the last inequality is from the fact that $u_2(2) \leq 0$. Also since $u_3(3)$ goes to $-\eta r$ when $\delta_3$ goes to 0, there exists $\tau_2$ such that for $\delta_3 \leq \tau_2$:

$$u_3(3) = -\eta \left( r\sqrt{1 + \delta_3^2} - \delta_3 u_2(2) \right) \geq -2\eta r.$$

Further,

$$u_3(3) - \delta_3 u_3(2) = -\eta \left( r\sqrt{1 + \delta_3^2} - \delta_3 u_2(2) \right) - \delta_3 \left( -\sigma + \eta \delta_3 \left( r\sqrt{1 + \delta_3^2} - \delta_3 u_2(2) \right) \right).$$

Again, this term goes to something strictly negative as $\delta_3$ goes to 0. This means that there exists $\tau_3$ such that for every $\delta_3 \leq \tau_3$ it holds that $u_3(3) - \delta_3 u_3(2) < 0$. Choosing $\delta_3 = \min\{\tau_1, \tau_2, \tau_3, 1\}$ concludes $u_3$. We will now calculate $u_4$. From what we have shown:

$$\nabla g(u_3) = -\delta_4 u_3(3)(e_4 - \delta_4 e_3),$$

which gives:

$$u_{3+1/2} = u_3 + \frac{r \nabla g(u_3)}{\|\nabla g(u_3)\|} = u_3 + \frac{r \delta_4}{\delta_4 \sqrt{1 + \delta_4^2}} e_4 - \frac{r \delta_4^2}{\delta_4 \sqrt{1 + \delta_4^2}} e_3$$

$$= u_3 + \frac{r}{\sqrt{1 + \delta_4^2}} e_4 - \frac{r \delta_4}{\sqrt{1 + \delta_4^2}} e_3.$$

Thus,

$$\nabla g(u_{3+1/2}) = (u_{3+1/2}(4) - \delta_4 u_{3+1/2}(3)) e_4 - \delta_4 (u_{3+1/2}(4) - \delta_4 u_{3+1/2}(3)) e_3$$

$$= \left( \frac{r}{\sqrt{1 + \delta_4^2}} + \frac{r \delta_4^2}{\sqrt{1 + \delta_4^2}} - \delta_4 u_3(3) \right) e_4 - \delta_4 \left( \frac{r}{\sqrt{1 + \delta_4^2}} + \frac{r \delta_4^2}{\sqrt{1 + \delta_4^2}} - \delta_4 u_3(3) \right) e_3$$

$$= \left( r\sqrt{1 + \delta_4^2} - \delta_4 u_3(3) \right) e_4 - \delta_4 \left( r\sqrt{1 + \delta_4^2} - \delta_4 u_3(3) \right) e_3.$$

Finally,

$$u_4 = u_3 - \eta \left( r\sqrt{1 + \delta_4^2} - \delta_4 u_3(3) \right) e_4 + \eta \delta_4 \left( r\sqrt{1 + \delta_4^2} - \delta_4 u_3(3) \right) e_3.$$

This gives:

$$-2\eta r \leq u_4(3) = u_3(3) + \eta \delta_4 \left( r\sqrt{1 + \delta_4^2} - \delta_4 u_3(3) \right).$$

Importantly $u_3(3)$ does not depend on $\delta_4$ so this term goes to $u_3(3) < -\eta r$ as $\delta_4$ goes to 0. This means that there exists $\theta_1$ such that for every $\delta_4 \leq \theta_1$ we have that $u_4(3) < -\frac{1}{2}\eta r$. Furthermore,

$$u_4(4) = -\eta \left( r\sqrt{1 + \delta_4^2} - \delta_4 u_3(3) \right) \leq -\eta r + \eta \delta_4 u_3(3) \leq -\eta r,$$

where the last inequality is from the fact that $u_3(3) \leq 0$. Also since $u_4(4)$ goes to $-\eta r$ when $\delta_4$ goes to 0, there exists $\theta_2$ such that for $\delta_4 \leq \theta_2$:

$$u_4(4) = -\eta \left( r\sqrt{1 + \delta_4^2} - \delta_4 u_3(3) \right) \geq -2\eta r.$$

Further,

$$u_4(4) - \delta_4 u_4(3) = -\eta \left( r\sqrt{1 + \delta_4^2} - \delta_4 u_3(3) \right) - \delta_4 \left( -u_3(3) + \eta \delta_4 \left( r\sqrt{1 + \delta_4^2} - \delta_4 u_3(3) \right) \right).$$

Again, this term goes to something strictly negative as $\delta_4$ goes to 0. This means that there exists $\theta_3$ such that for every $\delta_4 \leq \theta_3$ it holds that $u_4(4) - \delta_4 u_4(3) < 0$. Choosing $\delta_4 = \min\{\theta_1, \theta_2, \theta_3, 1\}$ concludes $u_4$ and the base case.

**Inductive step.** Assume this holds for $t' \leq t$. Notice that from the claim it holds that $u_{t'}$ does not depend on $\delta_t$ for $t' \leq t$. So we can choose $\delta_t$ now using $\{u_{t'}\}_{t' \leq t}$. We will calculate the SAM update for from $u_{t-1}$ to $u_t$ using the inductive assumption:

$$\nabla g(u_{t-1}) = -\delta_t u_t(t)(e_t - \delta_t e_{t-1})$$

which gives:

$$u_{t-1+1/2} = u_{t-1} + \frac{r \nabla g(u_{t-1})}{\|\nabla g(u_{t-1})\|} = u_{t-1} + \frac{r\delta_t}{\delta_t \sqrt{1 + \delta_t^2}} e_t - \frac{r\delta_t^2}{\delta_t \sqrt{1 + \delta_t^2}} e_{t-1}$$

$$= u_{t-1} + \frac{r}{\sqrt{1 + \delta_t^2}} e_t - \frac{r\delta_t}{\sqrt{1 + \delta_t^2}} e_{t-1}.$$

Thus,

$$\nabla g(u_{t-1+1/2}) = (u_{t-1+1/2}(t) - \delta_t u_{t-1+1/2}(t-1))(e_t - \delta_t e_{t-1})$$

$$= \left( \frac{r}{\sqrt{1 + \delta_t^2}} + \frac{r\delta_t^2}{\sqrt{1 + \delta_t^2}} - \delta_t u_{t-1}(t-1) \right)(e_t - \delta_t e_{t-1})$$

$$= \left( r\sqrt{1 + \delta_t^2} - \delta_t u_{t-1}(t-1) \right) e_t - \delta_t \left( r\sqrt{1 + \delta_t^2} - \delta_t u_{t-1}(t-1) \right) e_{t-1}.$$

Finally,

$$u_t = u_{t-1} - \eta \left( r\sqrt{1 + \delta_t^2} - \delta_t u_{t-1}(t-1) \right) e_t + \eta \delta_t \left( r\sqrt{1 + \delta_t^2} - \delta_t u_3 t - 1(t-1) \right) e_{t-1}.$$

This gives:

$$-2\eta r \leq u_t(t-1) = u_{t-1}(t-1) + \eta \delta_t \left( r\sqrt{1 + \delta_t^2} - \delta_t u_{t-1}(t-1) \right).$$

Importantly $u_{t-1}(t-1)$ does not depend on $\delta_t$ so this term goes to $u_{t-1}(t-1) < -\eta r$ as $\delta_t$ goes to 0. This means that there exists $\theta_1$ such that for every $\delta_t \leq \theta_1$ we have that $u_t(t-1) < -\frac{1}{2}\eta r$. Furthermore,

$$u_t(t) = -\eta \left( r\sqrt{1 + \delta_t^2} - \delta_t u_{t-1}(t-1) \right) \leq -\eta r + \eta \delta_t u_{t-1}(t-1) \leq -\eta r,$$

where the last inequality is from the fact that $u_{t-1}(t-1) \leq 0$. Also since $u_t(t)$ goes to $-\eta r$ when $\delta_4$ goes to 0, there exists $\theta_2$ such that for $\delta_t \leq \theta_2$:

$$u_t(t) = -\eta \left( r\sqrt{1 + \delta_t^2} - \delta_t u_{t-1}(t-1) \right) \geq -2\eta r.$$

Further,

$$u_t(t) - \delta_t u_t(t-1) =$$

$$- \eta \left( r\sqrt{1 + \delta_t^2} - \delta_t u_{t-1}(t-1) \right) - \delta_t \left( -u_{t-1}(t-1) + \eta \delta_t \left( r\sqrt{1 + \delta_t^2} - \delta_t u_{t-1}(t-1) \right) \right).$$

Again, this term goes to something strictly negative as $\delta_t$ goes to 0. This means that there exists $\theta_3$ such that for every $\delta_t \leq \theta_3$ it holds that $u_t(t) - \delta_t u_t(t-1) < 0$. Choosing $\delta_t = \min\{\theta_1, \theta_2, \theta_3\}$ concludes $u_t$ and the proof. $\square$

### C.4. Proof of Theorem 5.4

*Proof of Theorem 5.4.* The proof is identical to the proof of Theorem 4.4 except for using Theorem 5.1 instead of Theorem 4.1. □

