# OpenReview forum: "Flat Minima and Generalization: Insights from Stochastic Convex Optimization"
_ICML.cc/2026/Conference — ICML 2026 regular_

### Official Review · Reviewer_mVvr · 2026-02-15

**Soundness:** 4
**Presentation:** 4
**Significance:** 4
**Originality:** 3
**Overall Recommendation:** 5
**Confidence:** 4

**Summary:**

The paper challenges the “flat minima generalize better” narrative by constructing counterexamples in stochastic convex optimization with non-negative, $\beta$-smooth objectives. The main results are quoted in theorem forms (i) SA-ERM: even under a strong $\rho-flatness condition, flat empirical minimizers can have $\Omega(1)$ population risk while sharp minimizers generalize optimally. (ii) SA-GD: quite interestingly, training converges to flat minima at rate $\mathcal{O}(1/T)$ but population risk can be $\Omega(\eta^2(r-\rho)^2T)$ when the perturbation radius exceeds the flatness radius, with matching upper bounds via stability. (iii) SAM: minimizes empirical risk but may converge to sharp minima anyway, also with $\Omega(1)$ population risk, plus comparable upper bounds.

**Compliance With Llm Reviewing Policy:**

Affirmed.

**Key Questions For Authors:**

The paper overall is a very clear theoretical paper with great logic flow. Although here are a few points for the authors to consider.

1. The main limitation — which the authors acknowledge — is that all results are worst-case constructions in the convex setting. The constructions require dimension exponential in sample size ($d = 2n+1$ in Theorem 3.1), which is relevant to overparameterized settings but makes the counterexamples slightly artificial and deliberate. A natural follow-up question is whether these phenomena are pathological or generic — do they arise with positive probability over more generic settings, or only on specifically adversarial constructions?

2. Being a pure theory paper,  a simple numerical illustration of the separation on a concrete convex problem would help readers build intuition. Showing that SA-GD actually converges to a flat-but-poorly-generalizing minimum on a specific problem instance (even in 2D) would be valuable.

3. The strong flatness condition (Definition 2.1, requiring a perfectly flat region of radius $\rho$) is indeed very strong, as the authors note. Real loss landscapes are unlikely to have exactly flat regions. The results would be more compelling with a relaxed condition, such as some region where loss increase within $\epsilon$ of radius $\rho$. It would be nice to at least discuss this case.

4. The critical regime for the lower bounds is $r > \rho$ (perturbation radius exceeds true flatness radius). The paper leaves the $r \leq \rho$  regime as an open question. This is precisely the regime where a practitioner would hope SAM works (setting the perturbation radius conservatively), so understanding this case is important for practical implications. It would be great to at least include some calculations/discussions on this point.

5. The convex setting counterexample makes the results stronger in one sense (counterexamples even in the “easy” case), it limits direct relevance to deep learning where non-convexity is the primary challenge. The paper would benefit from discussing whether these convex counterexamples have analogues in shallow non-convex settings, for example, one can try to analyze what happens in the case of a two-layer networks.

**Limitations:**

Yes.

**Strengths And Weaknesses:**

This paper asks exactly the right question and provides rigorous, clearly stated answers. The constructed convex counterexamples is valuable, the separation between SA-GD/SAM and vanilla GD/SGD — showing that sharpness-aware methods can generalize strictly worse in settings where GD/SGD are optimal — is the paper’s sharpest result and is cleanly established.

The paper’s structure is very logical: first show flat ERM doesn’t help (Theorem 3.1), then show SA-GD finds flat minima but they generalize poorly (Theorems 4.1-4.4), then show SAM doesn’t even necessarily find flat minima (Theorems 5.1-5.4). Each result builds on and sharpens the previous one. The matching upper bounds via algorithmic stability provide a complete picture. The authors also admit to the limitations and gaps of their paper specifically in their limitation section.

---

> ### Author Rebuttal · Authors · 2026-03-30
>
> Thank you for your review and feedback.
>
> >”A natural follow-up question is whether these phenomena are pathological or generic — do they arise with positive probability over more generic settings, or only on specifically adversarial constructions?”
>
> Indeed, our main counterexamples are based on specific constructions but we note that this is always the case with worst-case lower bounds. While such constructions may appear specialized, they serve to characterize the limitations of the guarantees one can expect: in particular, they show that **flatness alone cannot universally ensure generalization without additional assumptions** (even in the controlled setting of smooth SCO).
>
> In this sense, our results do not rely on how common such instances are in **practice**, but rather demonstrate that any general **theory** linking flatness to generalization must depend on further assumptions. As you noted, an interesting direction for future work is to identify settings that rule out this behavior, or to consider more general distributions over structures, under which this phenomenon occurs only with small probability.
>
> >”Being a pure theory paper, a simple numerical illustration of the separation on a concrete convex problem would help readers build intuition. Showing that SA-GD actually converges to a flat-but-poorly-generalizing minimum on a specific problem instance (even in 2D) would be valuable.”
>
> Note that in low dimensions such as d = 2, any empirical minimum will generalize well due to uniform convergence guarantees. As a result, our lower bound (and any lower bound in SCO, for that matter) relies on a high-dimensional construction that does not admit a meaningful low-dimensional instantiation. To further help readers build intuition, we will expand the proof sketch with additional geometric explanations and visual diagrams to better convey the underlying mechanism.
>
> >”The strong flatness condition (Definition 2.1) is indeed very strong, as the authors note. Real loss landscapes are unlikely to have exactly flat regions. The results would be more compelling with a relaxed condition, such as some region where loss increase within $\epsilon$ of radius $\rho$ . It would be nice to at least discuss this case.”
>
> Our notion of flatness is indeed stronger than many commonly used definitions, like the one you mentioned. Importantly, and as discussed in the paper, these weaker notions are subsumed by our definition, in the sense that our lower bounds apply to them as well. We will emphasize this point more explicitly in the next version of the paper.
>
> >”The critical regime for the lower bounds is $r>\rho$ (perturbation radius exceeds true flatness radius). The paper leaves the $r\leq\rho$ regime as an open question. This is precisely the regime where a practitioner would hope SAM works (setting the perturbation radius conservatively)”
>
> The goal of those lower bounds is to show that the flat minimum that is found by common sharpness-aware algorithms may generalize poorly. You are correct that this phenomenon occurs in the regime where the perturbation radius exceeds the true flatness. We note, however, that even if $r$ only slightly overshoots the unknown true flatness (i.e., $r \simeq \rho + \frac{1}{\sqrt{T}}$), the algorithm can still optimize the empirical loss, while the population loss may be as large as $\Omega(1)$.
>
> As you point out, the regime in which the perturbation radius is smaller than the true flatness is also of interest, and we leave its investigation for future work.
>
> >”The convex setting counterexample makes the results stronger in one sense (counterexamples even in the “easy” case), it limits direct relevance to deep learning where non-convexity is the primary challenge. The paper would benefit from discussing whether these convex counterexamples have analogues in shallow non-convex settings, for example, one can try to analyze what happens in the case of a two-layer networks.”
>
> Thank you for this comment. Including such a discussion in the paper is indeed a very good idea, and we will incorporate one in the revision. In fact, our results indicate that from a generalization (as opposed to optimization) perspective, a fundamental barrier is already encountered in the convex case. The convex case further allows us to isolate different challenges and focus exclusively on generalization; indeed, it is considerably easier to demonstrate cases of failure in non-convex settings simply due to not converging to the global minimum.
>
> Regarding adapting our results to (two-layer) neural networks: we agree that this is also an interesting direction for future independent exploration. However, we also believe that the fundamental convex case deserves a focused treatment as a basis for exploration of more specialized settings, and this was our main goal in the current paper.

---

> > ### Author Rebuttal · Reviewer_mVvr · 2026-04-02
> >
> > Thanks to the authors for replying to the comments, look forward to seeing further work on this.

---

### Official Review · Reviewer_Wvt2 · 2026-03-03

**Soundness:** 3
**Presentation:** 3
**Significance:** 4
**Originality:** 3
**Overall Recommendation:** 5
**Confidence:** 2

**Summary:**

The paper studies the question of whether generic and algorithmically-biased flat minima have good generalization performance on stochastic convex optimization problems where the objective function is smooth. Theoretical evidence conclusively show that both types of flat minima do not always lead to better generalization performance compared to the best sharp minima. Interestingly, the results hold also for computationally efficient methods in sharpness-aware minimization.

**Compliance With Llm Reviewing Policy:**

Affirmed.

**Final Justification:**

The rebuttal fully addressed my concerns.

**Key Questions For Authors:**

1. Why is reducing the dimension from $d=2^n+1$ to polynomial in $n$ in Theorem 3.1 an interesting open question? I thought that this $d$ indicates the number of trainable parameters in an over-parameterized model, and indeed there are cases of deep neural networks where the dimension is exponentially larger than the number of training samples.

2. Is it correct that if a weaker definition of flat minima is used, then the sharp minima in your paper, especially the ones that generalize well, may become flat minima by a weaker definition? For example, in definition (ii) in Theorem 3.1, a ``mostly flat'' minimum might actually be considered sharp under the definition (ii). Here, mostly flat refers to the fact that the set of pertubations $v$ with high risks have measure zero.

**Limitations:**

Yes

**Strengths And Weaknesses:**

Strengths: The paper is technically sound, well-motivated. The theoretical contributions are significant.

Weaknesses: the setting is highly idealized. While this is already acknowledged by the authors, it does potentially mislead the readers because the definition of flat minima is very strong (see the question 2 below). Overall, I suggest the authors clarify this in the next version of the paper. For example, changing flat minima to ``strongly flat minima'' everywhere.

---

> ### Author Rebuttal · Authors · 2026-03-30
>
> Thank you for your review and feedback.
>
> >“the definition of flat minima is very strong. Overall, I suggest the authors clarify this in the next version of the paper. For example, changing flat minima to ``strongly flat minima'' everywhere.”
>
> Indeed, our notion of flatness is stronger than some of the commonly used definitions (and consequently, as emphasized in the paper, our lower bounds apply to them as well.)
>
> Regarding the phrasing, we appreciate the suggestion and will take it into account in the next version of the paper.
>
> >”Why is reducing the dimension from $d=2^n+1$ to polynomial in $n$ in Theorem 3.1 an interesting open question? I thought that this $d$  indicates the number of trainable parameters in an over-parameterized model, and indeed there are cases of deep neural networks where the dimension is exponentially larger than the number of training samples.”
>
> The dimension $d$ in our setting indeed indicates the number of parameters.
> In principle, we would like to exhibit the existence of a flat minimum that generalizes poorly in the smallest possible dimension. Since current upper bounds (for any ERM) suggest that the smallest dimension might be polynomial, even linear, it remains an interesting question to bridge the gap between these bounds. We will add a broader discussion on the dimension dependence of our results in the next revision of the paper.
>
> >”Is it correct that if a weaker definition of flat minima is used, then the sharp minima in your paper, especially the ones that generalize well, may become flat minima by a weaker definition? For example, in definition (ii) in Theorem 3.1, a ``mostly flat'' minimum might actually be considered sharp under the definition (ii). Here, mostly flat refers to the fact that the set of pertubations  with high risks have measure zero.”
>
> Thank you for this comment. The way the theorem is currently phrased, you are correct that the set of perturbations for which condition (ii) holds has measure zero. However, this can be easily addressed by increasing the dimension by a factor of two and slightly modifying the construction so that, with constant probability over the perturbation, the risk remains high. We will address this point in the next revision of the paper.

---

> > ### Author Rebuttal · Reviewer_Wvt2 · 2026-04-01
> >
> > Thanks for the answers. I'll keep the score of 5 (Accept).

---

### Official Review · Reviewer_jjyF · 2026-03-06

**Soundness:** 3
**Presentation:** 3
**Significance:** 1
**Originality:** 3
**Overall Recommendation:** 3
**Confidence:** 4

**Summary:**

This paper provides a negative result against the common belief that flat minima generalize well. They study the relation between flat minima and generalization in the setting of SCO. They show this negative result for both SA-GD and SAM proving that in this simple setup, although it converges to a flat minima, the population risk there has a constant lower bound.

**Compliance With Llm Reviewing Policy:**

Affirmed.

**Final Justification:**

I am still not convinced if SCO is the correct setup to study this distinction. The construction underlying the negative result feels somewhat tailored and maynot be what occurs in practice. However, I think the analysis with the current setup is correct. I will maintain my rating but leave it to the AC to make a final judgement.

**Key Questions For Authors:**

See weakness 1 and 4.

**Limitations:**

See weakness 4. Also the negative result holds only for a specific instance of an SCO problem,

**Strengths And Weaknesses:**

Strenghts:

1) While the flatness-generalization connection is widely invoked, this paper provides a sharpn separation between their generalization performation giving a clean negative results.

2) The paper is well written and the theorems are clear. The result is original and alhtough this is a negative result, the work is novel.

Weakness:

1) The result is a worst case lower bound constructed to a specific SCO problems. The counterexamples rely on carefully designed convex loss functions and data distributions where the empirical objective contains large flat valleys that hide directions affecting population risk. This might hinder the impact of the result as how common is such a distribution and problem instance in real deep learning problems. Its quite common to cook up a negative result using a loss function but the connection to practice is missing.

2) This brings to my second point which is lack of any experiments. Not even in the setting which the paper proposed its result, let alone a deep learning setting. For example, it is unclear how common are such constructions, whether it requires structured data distribution or similar failure modes also arrive in deep learning models.

3) Lack of intuition in the non convex setup is a major drawback. Since, most of the empirical success of SAM is in the non-convex. This raises the question of whether the analysis reflects the practical reason of success. Probably matrix factorizaiton can serve as a better setup to show the dominance of SAM over the SCO problem.

---

> ### Author Rebuttal · Authors · 2026-03-30
>
> Thank you for your review and feedback.
>
> >”The result is a worst case lower bound constructed to a specific SCO problems. [...] This might hinder the impact of the result as how common is such a distribution and problem instance in real deep learning problems.”
>
> Indeed, our main counterexamples are based on specific constructions but we note that this is always the case with worst-case lower bounds. While such constructions may appear specialized, they serve to characterize the limitations of the guarantees one can expect: in particular, they show that **flatness alone cannot universally ensure generalization without additional assumptions** (even in the controlled setting of convex smooth optimization).
>
> In this sense, our results do not rely on how common such instances are in **practice**, but rather demonstrate that any general **theory** linking flatness to generalization must depend on further structural or distributional assumptions. Identifying such assumptions in realistic settings could be an interesting direction for future work.
>
>
> >”lack of any experiments”
>
> The goal of this work is to study, from a theoretical perspective, the generalization behavior of algorithms that explicitly aim to find flat solutions in a fundamental optimization setting (SCO).
>
> >”Lack of intuition in the non convex setup is a major drawback [...] This raises the question of whether the analysis reflects the practical reason of success. Probably matrix factorizaiton can serve as a better setup to show the dominance of SAM over the SCO problem.”
>
> SCO is a fundamental setting for theoretically studying generalization phenomena, which was extensively studied over the last decade (e.g., Shalev-Shwartz et al., 2010; Feldman, 2016; Amir et al., 2021; Livni, 2024). While theoretical results do not always offer immediate practical guidance, their goal is to improve our understanding of the relationship between flatness and generalization and they provide insight into the behavior of generalization in a simple, fundamental setup.  Such an understanding is a prerequisite for understanding more involved scenarios, such as those encountered in practical settings.
>
>
> Regarding the non-convex setting, since our main results are lower bounds, imposing a favorable assumption such as convexity only strengthens them (“convex optimization” is contained in “non-convex optimization”). As for more specific non-convex settings, such as the matrix factorization you suggested, it would be an interesting independent direction for future exploration of similar questions.

---

> > ### Author Rebuttal · Reviewer_jjyF · 2026-04-02
> >
> > Thank you for the response. While I appreciate the technical contribution, I remain unconvinced that the SCO setup is sufficiently expressive to capture the phenomena typically associated with flat/sharp minima or implicit bias.
> >
> > As such, the motivation referring to “heavily overparameterized deep neural networks” appears to overstate the scope of the results, since the core mechanisms driving generalization in those settings are not present here.
> >
> > More broadly, the construction underlying the negative result feels somewhat tailored to demonstrate a separation rather than arising from a natural learning scenario.
> >
> > This does not invalidate the result, but it limits its applicability. I do not take issue with the correctness of the analysis; rather, I believe the overall message is stronger than what the setting justifies.

---

> > > ### Author Response · Authors · 2026-04-03
> > >
> > > Thank you for following up. You brought up some very good points - we will clarify them in more detail in the final version of the paper.
> > >
> > > > "the motivation referring to “heavily overparameterized deep neural networks” appears to overstate the scope of the results, since the core mechanisms driving generalization in those settings are not present here"
> > >
> > > Heavily overparameterized deep NN are indeed mentioned in our introduction as a general motivation for studying the connections between sharpness and generalization, while the technical results themselves pertain specifically to the SCO setting. We would argue that some of the “core mechanisms driving generalization” in overparameterized ML are in fact already exhibited in SCO, which is a primary motivation to the long line of theoretical research in SCO to which our paper belongs.
> > >
> > > In more detail: a primary challenge in modern learning theory, at least from the perspective of generalization, is the large number of parameters relative to the number of training samples; in such settings, there are many empirical minimizers, some of which generalize well while others generalize poorly, making generalization a central challenge. **The SCO setting is known to precisely exhibit this phenomenon** [1] and has therefore become a central focus in learning theory over the past decade, particularly in venues such as ICML and NeurIPS (e.g., [2]; [3]; [4]; [5]). Our work follows this line of research: we study a tractable setting that captures one important aspect of modern overparameterized learning, with the hope that such analysis can eventually inform our understanding of more practical models, including overparameterized deep neural networks.
> > >
> > >
> > > > "the construction underlying the negative result feels somewhat tailored to demonstrate a separation rather than arising from a natural learning scenario"
> > >
> > > Indeed, this a common (and essentially unavoidable) "inconvenience" with worst case analysis - the counterexamples are almost never deemed as "natural". However, recall that **the primary purpose of lower bounds is to rule out positive results in the same theoretical model** (in our case, smooth SCO) rather than to exhibit natural/practical cases of failure. Our lower bounds do precisely that: they rule out positive generalization/convergence results for SAM etc, even in the "easy" smooth (even realizable) SCO setting. We believe this result would be of interest to researchers working to improve methods and theoretical analyses of sharpness aware methods.
> > >
> > > > "I believe the overall message is stronger than what the setting justifies"
> > >
> > > By no means we intended to oversell our contribution, and we tried to be very careful and be fully transparent that the actual results apply in the specific model of smooth SCO. That being said, we will revisit our exposition and tone down the message wherever appropriate.
> > >
> > >
> > >
> > > [1] Shalev-Shwartz, S., Shamir, O., Srebro, N., and Sridharan, K. Learnability, stability and uniform convergence. The Journal of Machine Learning Research, 11:2635–2670,
> > > 2010.
> > >
> > > [2] Feldman, V. Generalization of ERM in stochastic convex optimization: The dimension strikes back. In Advances in Neural Information Processing Systems, volume 29, 2016.
> > >
> > > [3] Amir, I., Koren, T., and Livni, R. SGD generalizes better than gd (and regularization doesn’t help). In Conference on Learning Theory, pp. 63–92. PMLR, 2021.
> > >
> > > [4] Livni, Roi. "The sample complexity of gradient descent in stochastic convex optimization." Advances in Neural Information Processing Systems 37 (2024): 64215-64241.
> > >
> > > [5] Attias, Idan, et al. "Information complexity of stochastic convex optimization: Applications to generalization, memorization, and tracing." Forty-first International Conference on Machine Learning. 2024.

---

### Official Review · Reviewer_RTKi · 2026-03-17

**Soundness:** 4
**Presentation:** 3
**Significance:** 2
**Originality:** 3
**Overall Recommendation:** 4
**Confidence:** 3

**Summary:**

The paper demonstrates that flat minima, including flat minima found by some common flatness-seeking optimization algorithms, do not always represent solutions with good generalization. The paper shows this through several counterexamples in the setting of stochastic convex optimization, where the per-example loss is convex and the gradient Lipschitz continuous.

**Compliance With Llm Reviewing Policy:**

Affirmed.

**Final Justification:**

I believe the worst-case lower bounds are impressive and interesting in their own right, but I worry that the framing is too strong. For that reason, I’m satisfied with the determination of 4 (weak accept) rather than 5 (accept).

**Key Questions For Authors:**

The worst-case lower bounds depend on the size of the perturbation step r. The results assume constant r. Is leaving r constant common practice? Does your work suggest that annealing r could be an improvement on common practice?

The results essentially show that SA-GD and SAM don’t provide generalization guarantees for convex functions. Was it ever reasonable to expect that they did, given failures in other settings?

**Limitations:**

Generally yes. Perhaps directly commenting on how the results depend on the size of r would improve the paper.

**Strengths And Weaknesses:**

**Strengths**

The results appear to be correct and not obvious, the setting addressed is important, the statement of the results and their respective proofs (and additional proof sketch for Theorem 3.1) are clear.

**Weaknesses**

1 - The authors ascribe importance to the fact that their counterexamples are convex L-smooth functions. They argue that “already” having counterexamples in such a “fundamental” setting strengthens the larger argument about flatness not signifying generalization. Couldn’t one argue the opposite: the counterexamples are special in this way, perhaps suggesting that they are of less concern in practice? The paper could do more to justify the intuition that convex functions is a setting where sharpness-aware methods would be surprising to fail. The exponential dimensionality of the counterexample functions also suggests at least the possibility that one could avoid such bad minima in practice. Though possibly interesting on other terms, this work is somewhat disconnected from the motivating question of why overparametrized NNs work well in practice.

2 - There is some confusing language throughout. In particular, when describing flat minima with bad generalization found by SA-ERM, the paper uses “generic” and “arbitrary” in ways at odds with their usual meaning in mathematical contexts, where arbitrary does the same job as “any” or “for all” and “generic” means “almost surely” or at least typical in some sense. This is particularly confusing as it implies a different (and obviously much stronger) result than the one actually shown. For example the first sentence of Section 3 says, “… showing that an arbitrary minimizer of the SAER may exhibit a trivial \Omega(1) population risk,” when it probably intended to say, “… showing that some minimizers of the SAER exhibit a non-vanishing \Omega(1) population risk.” Note also the confusing use of “trivial” when something more near the opposite was likely intended. I believe better care could be taken in wording the theorems. For example constructions similar to “For every x, let y. Then…” were probably meant to be “Given x, let y. Then for every x…” Also, some things (like w^*) are not defined near the theorem. These are smaller matters of readability.

3 - Numerical examples to illustrate the results would be a good addition.

---

> ### Author Rebuttal · Authors · 2026-03-30
>
> Thank you for your review and feedback.
>
> > “Couldn’t one argue the opposite: the counterexamples are special in this way, perhaps suggesting that they are of less concern in practice?”
>
> Indeed, our main counterexamples are based on specific constructions but we note that this is always the case with worst-case lower bounds. While such constructions may appear specialized, they serve to characterize the limitations of the guarantees one can expect: in particular, they show that **flatness alone cannot universally ensure generalization without additional assumptions** (even in the setting of smooth SCO).
>
> In this sense, our results do not rely on how common such instances are in **practice**, but rather demonstrate that any general **theory** linking flatness to generalization must depend on further assumptions. Identifying such assumptions in realistic settings could be an interesting direction for future work.
>
>
> > “The paper could do more to justify the intuition that convex functions is a setting where sharpness-aware methods would be surprising to fail”
>
> The goal of the paper is to study the flatness of minima as a condition for generalization. SCO is widely considered as a clean theoretical setting for establishing such results, and is home to many of the algorithms being used in ML today (SGD, Adagrad, etc). Since many well-known characterizations of well-generalizing minima, such as minimum norm or maximum margin, have been formally justified in SCO, we find it quite surprising that flatness, despite being considered a prominent indicator of generalization, fails to generalize even in this fundamental setting.
>
> >”The exponential dimensionality of the counterexample functions also suggests at least the possibility that one could avoid such bad minima in practice.”
>
> Continuing our first comment, our bounds do not impose any assumptions on the dimension. It is indeed an interesting question whether, under restrictions on the dimension (e.g., polynomial in the sample size), flat minima would exhibit improved generalization.
>
> >”Though possibly interesting on other terms, this work is somewhat disconnected from the motivating question of why overparametrized NNs work well in practice.”
>
> This paper is a theoretical work aimed at analyzing the relationship between flat minima and generalization in a basic setting, as discussed in our second comment. It would be interesting in future work to investigate whether additional structural assumptions, such as specific neural network architectures, lead to different conclusions.
> While our theoretical results do not immediately offer **practical guidance**, their goal is to improve **theoretical understanding** of the relationship between flatness and generalization and they provide insight into the behavior of generalization in a simple setup. Such an understanding is a prerequisite for understanding more involved scenarios, such as those encountered in practical settings.
>
> >”There is some confusing language throughout”
>
> Many thanks for your suggestions, they will help us to improve the clarity of the paper. We will revise the phrasing accordingly in the next revision.
>
> >”Numerical examples to illustrate the results would be a good addition.”
>
> Again, this is a theory-focused paper the goal of which is to theoretically study the generalization behavior of algorithms that explicitly aim to find flat solutions in a fundamental optimization setting (SCO).
>
> >”The worst-case lower bounds depend on the size of the perturbation step r.”
>
> Our lower bounds for SA-GD and SAM depend on $r$. We actually see this as a nice property of our results. This shows that, when $r = 0$ (and both methods reduce to standard GD), the generalization performance is optimal in this setting, and increasing $r$ (for actively looking for a flat solution) hurts the performance of the algorithm.
>
> >”The results assume constant r. Is leaving r constant common practice?”
>
> We emphasize that we **do not** assume a constant $r$ anywhere in the paper. Our lower bounds show that SA-GD and SAM can incur population loss as large as $\Omega(1)$ even when the gap between $r$ and the true flatness $\rho$ is small (order of $\frac{1}{\sqrt{T}}$). Regarding practice, it has been argued that using a constant $r$ is common in practice (e.g., Si et al., 2023). However, our results do not rely on this assumption, and in fact, a constant $r$ would only further strengthen our lower bounds.
>
> >”The results essentially show that SA-GD and SAM don’t provide generalization guarantees for convex functions. Was it ever reasonable to expect that they did, given failures in other settings?”
>
> We emphasize that, to our knowledge, our theoretical lower bounds for SA-GD and SAM are the first in *any* setting. Prior works either show positive theoretical results for specialized settings (the strongly convex case) or discuss the empirical performance of these algorithms. We will explain this point better in the next revision.

---

> > ### Author Rebuttal · Reviewer_RTKi · 2026-04-08
> >
> > I appreciate the promise to correct the language. I want to emphasize that the non-standard usages of “generic” and “arbitrary” were serious for a paper whose contribution is a theoretical result. Contingent on such changes being made the paper merits the highest presentation rating.
> >
> > This work certainly shows that the link between generalization and flatness, to the extent it holds, depends on assumptions. But, crucially, the specialness of the setting and examples means this work says *less* about what those assumptions are, not *more*.
> >
> > I believe the worst-case lower bounds are impressive and interesting in their own right, but I worry that the framing is too strong. For that reason, I’m satisfied with the determination of 4 (weak accept) rather than 5 (accept).

---

### Decision · Program_Chairs · 2026-04-30

**Decision:**

Accept (regular)

**Comment:**

This manuscript studies the theoretical link between flat minima and generalization within the canonical setting of stochastic convex optimization. All reviewers find the topic interesting and the result sound. Multiple reviewers, including the reviewer Wvt2 who gave a 5, highlight that the framing can be too strong due to the specialized setting and raise the issue of overclaim, which I second. The authors should revise their language to explicitly clarify their reliance on a very strong definition of flatness. Furthermore, the paper should one down their claims regarding overparameterized deep neural networks to better match the limitations of their worst-case SCO setting, and include the promised geometric diagrams to help build reader intuition.